# Option-aware Temporally Abstracted Value for Offline Goal-Conditioned Reinforcement Learning

**Hongjoon Ahn**[1*] **Heewoong Choi**[1*] **Jisu Han**[2*] **Taesup Moon**[1,2,3†]

[1] Department of Electrical and Computer Engineering (ECE), Seoul National University
[2] Interdisciplinary Program in Artificial Intelligence (IPAI), Seoul National University
[3] ASRI / INMC, Seoul National University
{hong0805, chw0501, jshcdi, tsmoon}@snu.ac.kr

## Abstract

Offline goal-conditioned reinforcement learning (GCRL) offers a practical learning paradigm in which goal-reaching policies are trained from abundant state–action trajectory datasets without additional environment interaction. However, offline GCRL still struggles with long-horizon tasks, even with recent advances that employ hierarchical policy structures, such as HIQL [33]. Identifying the root cause of this challenge, we observe the following insight. Firstly, performance bottlenecks mainly stem from the high-level policy's inability to generate appropriate subgoals. Secondly, when learning the high-level policy in the long-horizon regime, the sign of the advantage estimate frequently becomes incorrect. Thus, we argue that improving the value function to produce a clear advantage estimate for learning the high-level policy is essential. In this paper, we propose a simple yet effective solution: ***Option-aware Temporally Abstracted*** value learning, dubbed **OTA**, which incorporates temporal abstraction into the temporal-difference learning process. By modifying the value update to be *option-aware*, our approach contracts the effective horizon length, enabling better advantage estimates even in long-horizon regimes. We experimentally show that the high-level policy learned using the OTA value function achieves strong performance on complex tasks from OGBench [32], a recently proposed offline GCRL benchmark, including maze navigation and visual robotic manipulation environments. Our code is available at https://github.com/ota-v/ota-v

## 1 Introduction

Offline goal-conditioned reinforcement learning (GCRL) has emerged as a practical framework for real-world applications by leveraging pre-collected datasets to train goal-reaching policies without requiring additional environment interaction [23, 32]. However, learning an accurate goal-conditioned value function in long-horizon settings remains a major challenge, as naively training the value function often leads to noisy estimates and erroneous policies [35, 20, 33]. To mitigate the learning of an erroneous policy, Hierarchical Implicit Q-Learning (HIQL) [33], one of the state-of-the-art methods, adopts a simple hierarchical structure in which a high-level policy predicts subgoals, and a low-level policy learns to execute actions toward those subgoals. Though a hierarchical policy is still learned from the noisy value function, both policies receive more reliable learning signals than when training a flat, non-hierarchical policy. However, despite reasonable performance gains of hierarchical methods in some long-horizon environments, a recent challenging benchmark [32]

---

*Equal Contribution.
†Corresponding author.

39th Conference on Neural Information Processing Systems (NeurIPS 2025).

reveals that such a hierarchical policy still cannot solve more complex tasks, such as long-horizon robotic locomotion or robotic manipulation.

To understand the failure in complex tasks more deeply, we raise the following question: *Low-level policy vs. high-level policy: which is the bottleneck of HIQL?* To answer this question, we analyze the hierarchical policy in failure cases by generating oracle subgoals for the low-level policy. Interestingly, we observe that the low-level policy achieves these subgoals with high accuracy, indicating that the failure stems from the inability of the high-level policy to generate appropriate subgoals. The limited performance primarily results from a noisy value function, which fails to provide sufficiently informative learning signals for effectively training the high-level policy in long-horizon scenarios.

Based on the phenomenon that the high-level policy eventually failed to extract meaningful learning signals from the value function, we identify the primary cause of these noisy signals as the *order inconsistency of the learned value function in the long-horizon setting*. Our analysis reveals that when the distance between the state and the goal exceeds a certain temporal horizon, the sign of the advantage estimate is incorrect, causing erroneous regression weights for learning the high-level policy. Considering the issue with the value function, we argue that designing a value function that can produce a clear advantage estimate for learning the high-level policy is necessary.

Motivated by the observation that the low-level policy performs remarkably well at reaching short-horizon subgoals, we propose a simple yet effective value function learning scheme for high-level policy learning that reduces the horizon between the state and the goal. Specifically, we leverage the notion of *option* [45], a temporally-extended course of action, by updating the value over sequences of primitive actions. This *option-aware* value learning substantially shortens the effective horizon compared to primitive action-aware value learning [21], mitigating errors in long-horizon value estimation. We evaluate our approach on maze and robotic visual manipulation tasks from OGBench [32], and empirically show that using our value function enables the high-level policy to achieve superior performance on long-horizon tasks.

In summary, our contributions are threefold:

- Through analysis of the failure cases of hierarchical policies, we identify that the failures stem from the inability of the high-level policy to generate appropriate subgoals. Furthermore, we observe that the value function used for high-level policy learning has significant errors when the distance between the state and the goal is large.

- To tackle this problem, we propose *Option-aware Temporally Abstracted (OTA)* value learning, which reduces the effective horizon compared to the conventional value learning objective [21].

- Our experiments show that, even across long state-to-goal horizons, our value function achieves significantly lower errors, enabling the hierarchical policy to successfully solve complex maze and robotic manipulation tasks.

## 2   Related Work

**GCRL.** GCRL aims to train goal-conditioned policies to reach *arbitrary* goal states from given initial states, rather than optimizing for a single, fixed task [42, 25]. Our work focuses specifically on offline GCRL [4, 26, 52, 33, 43, 32], in which goal-conditioned policies are learned entirely from pre-collected datasets without further environment interaction. Due to the sparse rewards in goal-reaching tasks, offline GCRL has relied on hindsight data relabeling [1, 40, 55], and more recently, imitation learning and value-based methods have been explored to better leverage suboptimal datasets [6, 11, 52, 12]. In these works, the value function is typically learned through temporal-difference (TD) methods [21, 34], or through alternative techniques such as state-occupancy matching [26, 7], contrastive learning [27, 9, 24], and quasimetric learning [50]. However, whether the value functions can effectively generalize to long-horizon tasks remains an open question [32].

**Hierarchical RL.** Achieving long-horizon goals remains a fundamental challenge in GCRL [37, 35, 20, 33]. To address this, hierarchical RL methods have adopted either graph-based planning in the state space [8, 16, 54, 20, 53, 19] or waypoint-based subgoal generation [5, 22, 47, 28, 17, 13, 31, 30, 18, 3, 33, 51]. However, graph-based planning methods incur high computational overhead and architectural complexity. Waypoint-based approaches also face challenges in generating effective

subgoals in long-horizon settings, due to inaccurate value estimates when the state is far from the goal.

**Option framework.** To enhance the planning capabilities of an agent over long time horizons, one effective approach is to leverage temporal abstraction through the option framework, which involves learning sub-policies known as *options* [14, 45, 41, 35]. In this framework, options serve as temporally extended actions that enable planning across multiple time scales. After establishing the theoretical connection between the option framework and semi-Markov decision processes [45], research has progressed toward end-to-end option learning [39, 44, 2, 46] and automatic option discovery [38, 15]. Our method is closely related to HIQL [33], which trains a high-level policy to generate subgoals. However, unlike HIQL, our approach leverages options defined in offline datasets to effectively reduce the planning horizon during value function training. As a result, our high-level policy can generate subgoals over longer temporal horizons without relying on explicit option learning or option discovery.

## 3 Preliminaries

**Problem setting.** Offline GCRL is defined over a Markov Decision Process (MDP), consisting of $(\mathcal{S}, \mathcal{A}, \mathcal{G}, r, \gamma, p_0, p)$ in which $\mathcal{S}$ denotes the state space, $\mathcal{A}$ the action space, $\mathcal{G}$ the goal space, $r(s, g)$ the goal-conditioned reward function for state $s \in \mathcal{S}$ and goal $g \in \mathcal{G}$, $\gamma$ the discount factor, $p_0(\cdot)$ the initial state distribution, and $p(\cdot|s, a)$ the environment transition dynamics for state $s \in \mathcal{S}$ and action $a \in \mathcal{A}$. We also denote $V(s, g)$ as the goal-conditioned value function at state $s$ given goal $g$. We assume that the goal space is the same as the state space (*i.e.*, $\mathcal{S} = \mathcal{G}$). An offline dataset $\mathcal{D}$ consists of trajectories $\tau = (s_0, a_0, s_1, \ldots, s_T)$, each sampled from an unknown behavior policy $\mu$. The objective is to learn an optimal goal-conditioned policy $\pi(a|s, g)$ that maximizes the expected cumulative return $\mathcal{J}(\pi) = E_{\tau \sim p^\pi(\tau), g \sim p(g)}[\sum_{t=0}^T \gamma^t r(s_t, g)]$, where $p^\pi(\tau) = p_0(s_0)\Pi_{t=0}^{T-1} p(s_{t+1}|s_t, a_t)\pi(a_t|s_t, g)$, and $p(g)$ is a goal distribution.

**Hierarchical Implicit Q-Learning (HIQL).** In GCRL, accurately estimating the value function for distant goals is the main challenge in solving complex long-horizon tasks [16, 20, 33]. To address this issue, HIQL [33] proposed a hierarchical policy structure that utilizes a value function learned with IQL [21]. This hierarchical design enables the agent to produce effective actions even when value estimates for distant goals are noisy or unreliable. More specifically, HIQL trains a goal-conditioned state-value function $V$ with the following loss:

$$\mathcal{L}(V) = \mathbb{E}_{(s,s') \sim \mathcal{D}, \, g \sim p(g)} \left[ L_2^\tau \left( r(s, g) + \gamma \bar{V}(s', g) - V(s, g) \right) \right], \tag{1}$$

where the expectile loss is defined as $L_2^\tau(u) = |\tau - \mathbf{1}(u < 0)|u^2$, with $\tau > 0.5$, and $\bar{V}$ denotes the target $V$ network.[3] Following prior works [1, 8, 3, 50, 33, 51], we adopt the sparse reward $r(s, g) = -\mathbf{1}\{s \neq g\}$. Under this reward, the optimal value $|V^\star(s, g)|$ corresponds to the *discounted temporal distance*, *i.e.*, a discounted measure of the minimum number of environment steps required to reach the goal $g$ from state $s$. HIQL separates policy extraction[4] into two levels: a high-level policy $\pi^h(s_{t+k}|s_t, g)$ generates a $k$-step subgoal to guide progress toward the goal, while a low-level policy $\pi^\ell(a_t|s_t, s_{t+k})$ produces primitive actions to reach the subgoal. Both policies are extracted using advantage-weighted regression (AWR) [48, 36, 29] with the following objective:

$$\mathcal{J}(\pi^h) = \mathbb{E}_{(s_t, s_{t+k}, g) \sim \mathcal{D}} \left[ \exp \left( \beta^h \cdot A^h(s_t, s_{t+k}, g) \right) \log \pi^h(s_{t+k}|s_t, g) \right], \tag{2}$$

$$\mathcal{J}(\pi^\ell) = \mathbb{E}_{(s_t, a_t, s_{t+1}, s_{t+k}) \sim \mathcal{D}} \left[ \exp \left( \beta^\ell \cdot A^\ell(s_t, s_{t+1}, s_{t+k}) \right) \log \pi^\ell(a_t|s_t, s_{t+k}) \right], \tag{3}$$

where $\beta^h$ and $\beta^l$ are inverse temperature parameters, $A^h(s_t, s_{t+k}, g) = V^h(s_{t+k}, g) - V^h(s_t, g)$ denotes the high-level policy advantage, and $A^\ell(s_t, s_{t+1}, s_{t+k}) = V^\ell(s_{t+1}, s_{t+k}) - V^\ell(s_t, s_{t+k})$ denotes the low-level policy advantage. HIQL uses a single goal-conditioned value function $V$, which is shared between both $\pi^h$ and $\pi^\ell$ (*i.e.*, $V^h = V^\ell = V$). However, despite this design, HIQL still struggles with long-horizon, complex tasks, as shown in the recent offline GCRL benchmark, OGBench [32].

---

[3]Note that since the inherent over-estimation problem of IQL, in this paper, we assume that the environment dynamics is deterministic.

[4]Policy extraction refers to learning a policy from a learned value function, emphasizing the separation between value learning and policy learning.

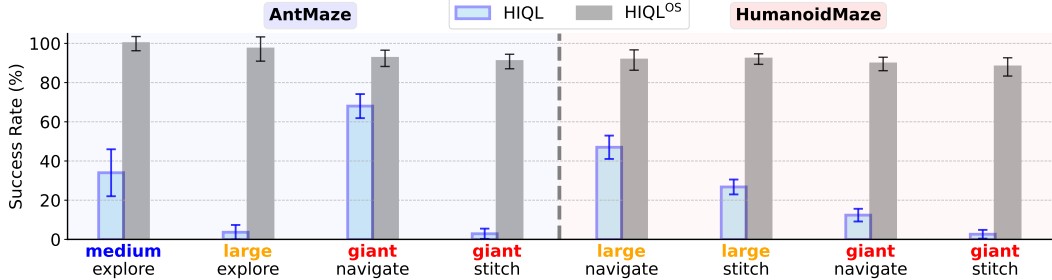

Figure 1: **High-level policy is problematic.** We evaluate HIQL by varying only the high-level policy while keeping the low-level policy fixed. The x-axis denotes different tasks under maze sizes and data types (See Section 6.1 for task details). Using learned high-level policy (HIQL, $\pi = \pi^{\ell} \circ \pi^{h}$), performance drops, whereas using the oracle high-level policy (HIQL$^{\mathrm{OS}}$, $\pi = \pi^{\ell} \circ \pi^{h}_{\mathrm{oracle}}$) achieves high success rates, indicating the high-level policy is the main bottleneck.

## 4 Motivation

### 4.1 Low-Level Policy vs. High-Level Policy: Which is the Bottleneck of HIQL?

We investigate the failure cases of HIQL in long-horizon scenarios by identifying whether the main performance bottleneck is in the low-level policy or the high-level policy. To examine this, we fix the low-level policy $\pi^{\ell}$ and replace the high-level policy $\pi^{h}$ with an oracle policy $\pi^{h}_{\mathrm{oracle}}$, which always provides optimal subgoals reachable within a short horizon.[5] We refer to this variant as HIQL$^{\mathrm{OS}}$, and pose the following hypothesis: if HIQL$^{\mathrm{OS}}$ still fails in long-horizon tasks, then the low-level policy struggles to reach short-horizon subgoals. Conversely, if it achieves a high success rate, the main problem lies in the high-level policy.

Figure 1 shows the results of HIQL and HIQL$^{\mathrm{OS}}$ on eight challenging maze navigation tasks from OGBench [32]. HIQL achieves less than 20% success rate on many tasks, indicating that HIQL significantly fails to solve the long-horizon tasks. In contrast, we note that HIQL$^{\mathrm{OS}}$ achieves a much higher success rate around 90%. These results indicate that, although the low-level policy generalizes well in short-horizon settings when provided with accurate subgoals, the primary failure of HIQL in long-horizon scenarios stems from inaccuracies in the high-level policy.

We identify two potential issues in Equation (2) that may underlie the failure of high-level policy learning: (1) an inadequate policy extraction scheme (*i.e.*, the regression component in Equation (2)), and (2) an inaccurately learned value function (*i.e.*, the advantage term in Equation (2)). Since the same policy extraction scheme enables successful low-level policy learning, we do not consider it to be the primary cause of failure. This suggests that the inaccurate value function used in the high-level policy advantage term may be the key contributor to the failure. In particular, as the distance between $s_t$ and $g$ increases, the value estimates become increasingly erroneous, leading to an imprecise evaluation of the high-level advantage. Although HIQL attempts to mitigate the noise in estimating the long-horizon value $V^h$ through its hierarchical structure, the high-level advantage may still be substantially erroneous. In the following subsection, we carefully analyze how such errors in estimating $V^h$ adversely affect high-level policy learning.

### 4.2 Order Inconsistency of the Learned Value Function in the Long-Horizon Setting

Before analyzing the learned $V^h$ in HIQL, we first define *order consistency* of the value function.

**Definition 4.1.** *(Order consistency) Assume that the environment is deterministic. Let $\tau^{\star} = (s_0, s_1, \ldots, s_T = g)$ denote the optimal trajectory induced by the optimal policy $\pi^{\star}(\cdot \mid s, g)$, from the initial state $s_0$ to the goal $g$, and let $V$ be a learned value function. Given $s_i, s_j \in \tau^{*}$ with $j > i$, we say that $V$ **satisfies order consistency with respect to** $(s_i, s_j, g)$ if and only if $V(s_j, g) > V(s_i, g)$.*

---

[5]Specifically, $\pi^{h}_{\mathrm{oracle}}$ provides as a subgoal the center of an adjacent maze cell that lies on the shortest path from the current state to the goal.

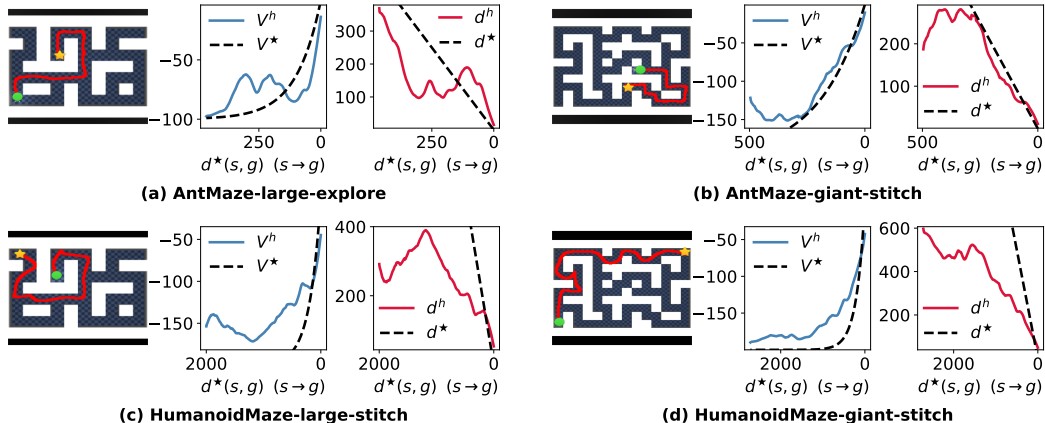

**(a) AntMaze-large-explore**

**(b) AntMaze-giant-stitch**

**(c) HumanoidMaze-large-stitch**

**(d) HumanoidMaze-giant-stitch**

Figure 2: **Value order inconsistency in long-horizon settings.** (*Left*) We collect optimal trajectories from the initial state (●) to the goal (⭐). (*Middle*) At each state along the trajectory, we compare the high-level value from HIQL ($V^h$) and the optimal ($V^\star$). (*Right*) To better illustrate value order consistency, we convert the values into temporal distances: HIQL ($d^h$) and the optimal ($d^\star$).

Consider an optimal trajectory $\tau^\star = \{s_0, s_1, \ldots, s_T\}$, generated by an oracle policy. Along this trajectory, the optimal value function is increasing, such that $V^\star(s_j, g) > V^\star(s_i, g)$ for all $j > i$. Thus, value order consistency refers to the alignment between the order induced by $V^h$ and that induced by $V^\star$. We argue that achieving the order consistency between $V(s_t, g)$ and $V(s_{t+k}, g)$ is critical, as sign mismatches can invert the high-level advantage estimate $A^h$ and hinder the learning of an appropriate high-level policy. With large $k$ values (*e.g.*, 25 in `AntMaze` 100 in `HumanoidMaze`), the sign mismatch of the advantage estimate can lead to significant performance degradation. When the advantage sign is incorrect, the magnitude of regression weights (which is the exponentiated advantage) is drastically increased or decreased, leading to improper subgoal regression for high-level policy. For example, if the range of an advantage is $[-1, 1]$ with $\beta^h = 3$, the regression weights vary from $e^{-3} \approx 0.05$ to $e^3 \approx 20$, indicating that a sign flip can significantly change the weight magnitude.

To check whether the learned $V^h$ of HIQL achieves the order consistency or not, we collected optimal trajectories for four different long-horizon tasks with specified goals using near-optimal policies, as illustrated in Figure 2. The trajectory lengths varied from 250 to 2000 steps. For each state $s_t$ in the trajectory, we then visualize the learned value $V^h(s_t, g)$ alongside the optimal value function, computed as $V^\star(s_t, g) = -\left(1 - \gamma^{d^\star(s_t, g)}\right) / (1 - \gamma)$, in which $d^\star(s_t, g)$ denotes the temporal distance between $s_t$ and $g$. Since the value decays exponentially as the distance to the goal increases due to the discount factor $\gamma$, directly comparing relative values is visually challenging. Hence, we transform $V^h(s, g)$ into estimated temporal distances using the following equation: $d^h(s, g) = \log\left(1 + (1 - \gamma)V^h(s, g)\right) / \log \gamma$. In this form, the criterion for value order consistency becomes $d^h(s_i, g) > d^h(s_j, g)$, where $j > i$.

As shown in Figure 2, we note that $V^h$ closely matches $V^\star$ when the state is near the goal (*i.e.*, $d^\star(s, g) \approx 0$). This alignment explains the strong performance of the low-level policy presented in Figure 1. However, when the state-goal distance exceeds a certain temporal horizon, the value order inconsistency frequently arises between $V^h(s_t, g)$ and $V^h(s_{t+k}, g)$ due to the non-monotonicity of the learned $V^h$.[6] This is due to the well-known fact that the learning target for the value in Equation (1) becomes noisier as the horizon becomes longer, and shows why the use of $V^h$ becomes less effective in high-level policy learning for long-horizon settings.

Motivated by the observation that $V^h$ aligns well with $V^\star$ and achieves order consistency in short-horizon settings, we propose a simple yet effective solution based on *temporal abstraction* [45]. This approach enables high-level value function learning to provide appropriate advantage estimates, even when $d^\star(s, g)$ is large.

---

[6]The hyperparameter $k$ in HIQL is chosen based on the characteristics of the environments and datasets. In Figure 2, $k = 25$ for the `AntMaze` task and $k = 100$ for the `HumanoidMaze` task.

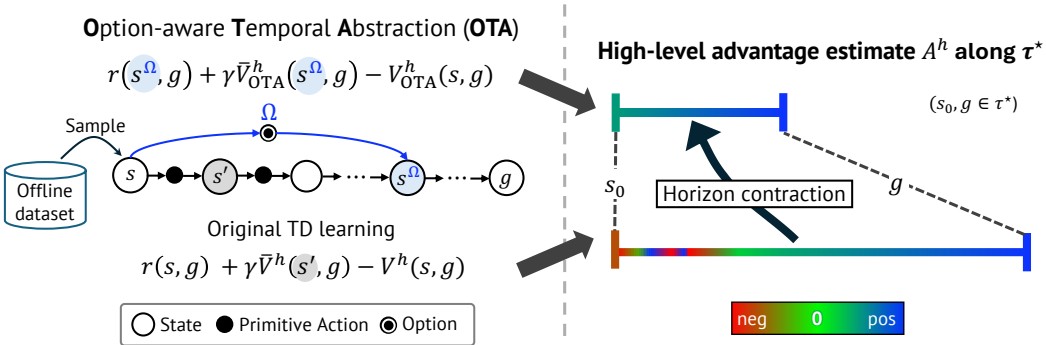

Figure 3: **Option-aware temporal abstraction.** (*Left*) OTA achieves temporal abstraction by computing the reward and target value from the state reached after executing the option (*i.e.*, $s^\Omega$). (*Right*) By leveraging temporal abstraction, OTA provides clear high-level advantage estimates, particularly in long-horizon tasks.

## 5 Option-aware Temporally Abstracted (OTA) Value

In this section, we propose a straightforward solution for learning $V^h(s,g)$ by leveraging *options* [45] to reduce the horizon length. An *option* can be regarded as a temporally extended sequence of primitive actions that enable temporal abstraction. In our offline RL setting, an option starting from the state $s_t$ corresponds to a sequence of $n$ actions $(a_t, a_{t+1}, \ldots, a_{t+n-1})$ extracted from trajectories in the offline dataset. By using temporally extended actions in planning, we reduce the *effective horizon length*, referring to the number of planning steps, to approximately $d^\star(s_t, g)/n$. Therefore, to ensure that the high-level value $V^h$ is suitable for long-term planning, we modify the reward and target value in Equation (1) to be *option-aware*.

Specifically, for a given abstraction factor $n$ and goal $g$, we define an option $\Omega_{n,g} = (\mathcal{I}, \mu, \beta_{n,g})$, where $\mathcal{I} = \mathcal{S}$ is the initiation set, $\mu$ is the behavior policy used to collect the offline dataset $\mathcal{D}$, and $\beta_{n,g}$ is a timeout-based termination condition that ends the option after $n$ steps or upon reaching $g$. Let $s'(\Omega_{n,g}, s_t)$ denote the state resulting from executing $\Omega_{n,g}$ at state $s_t$, which is either $s_{t+n}$ or $g$. For brevity, we denote $s'(\Omega_{n,g}, s)$ as $s^\Omega$. Then, we reformulate the value learning objective in Equation (1) into an *option-aware* variant:

$$\mathcal{L}(V^h_{\text{OTA}}, n) = \mathbb{E}_{(s,s^\Omega)\sim\mathcal{D}, g\sim p(g)}[L_2^\tau(r(s^\Omega, g) + \gamma \bar{V}^h_{\text{OTA}}(s^\Omega, g) - V^h_{\text{OTA}}(s, g))], \quad (4)$$

where $r(s^\Omega, g) = -\mathbf{1}\{s^\Omega \neq g\}$.[7] We refer to $V^h_{\text{OTA}}$ as the *Option-aware Temporally Abstracted* (OTA) value function.

We argue that the high-level value function $V^h_{\text{OTA}}$ would effectively address the value order inconsistency. Using a 1-step target for value learning tends to be more sensitive to noise, especially in long-horizon tasks, whereas an option-aware target mitigates noise and empirically produces more order-consistent value estimates. The overall framework for learning $V^h_{\text{OTA}}$ is illustrated in Figure 3.

**Connection to $n$-step TD learning.** The target used in $n$-step TD learning and that in OTA value learning are closely related, as both primarily rely on the $n$-step forward value.[8] However, the key distinction between $n$-step TD and OTA lies in the choice of the discount factor $\gamma$, which controls how information decays during the TD update. Standard $n$-step TD learning typically uses the same $\gamma$ as in 1-step TD, causing the discount factor applied to the $n$-step target to decay exponentially with $n$. In contrast, the discount factor in the OTA target is independent of $n$. This excessive decay in the standard $n$-step target hinders the order-consistent value learning, indicating that a direct extension from the $n$-step target to the OTA target is not straightforward. Instead, temporal abstraction through the option framework provides a natural explanation for the insights presented in Section 4.2. As shown empirically in 6.5, with respect to value order consistency, standard $n$-step TD learning offers no advantage over 1-step TD learning.

---

[7]We highlight the differences from Equation (1) in Equation (4).
[8]The TD target used in $n$-step TD learning is $\sum_{i=0}^{n-1} -\gamma^i \cdot \mathbf{1}(s_{t+i} \neq g) + \gamma^n V(s_{t+n}, g)$.

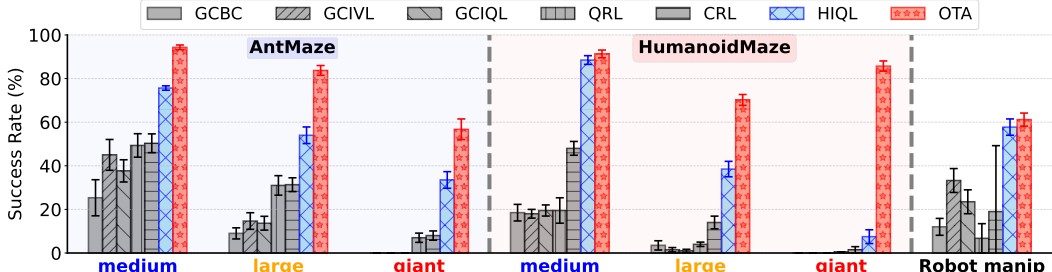

Figure 4: **Evaluation on OGBench.** We run 8 seeds for each dataset and use the performance reported in OGBench for the baselines. For maze tasks, we report the average success rate grouped by maze size. For visual robotic manipulation, we report the average success rate across the four tasks.

## 6 Experiments

### 6.1 Experiment Setup

**Tasks.** We use OGBench [32], a recently proposed offline GCRL benchmark designed for realistic environments, long-horizon scenarios, and multi-goal evaluation. The `Maze` environment consists of long-horizon navigation tasks that evaluate whether the agent can reach a specified goal position from a given initial state. The `Maze` environments are categorized by agent type (`PointMaze`, `AntMaze`, and `HumanoidMaze`), maze size (`medium`, `large`, and `giant`), and the type of trajectories in the dataset (`navigate`, `stitch`, and `explore`). The `Maze` environments are well suited to evaluating performance in long-horizon settings. For example, the `HumanoidMaze-giant` environment has a maximum episode length of 4000 steps.

The **Visual-cube** and **Visual-scene** environments focus on visual robotic manipulation tasks. In `Visual-cube`, the task involves manipulating and stacking cube blocks to reach a specified goal configuration. This environment is categorized by the number of cubes: `single`, `double`, and `triple`. In contrast, `Visual-scene` requires the agent to control everyday objects like windows, drawers, or two-button locks in a specific sequence. Both visual environments use high-dimensional, pixel-based observations with $64 \times 64 \times 3$ RGB images. The robotic manipulation environments have shorter episode lengths (250 to 1000 steps) compared to the `Maze` environments. These robotic environments are a strong benchmark for evaluating the performance of an algorithm on high-dimensional visual inputs. A detailed description of the environments is provided in Appendix B.1.

**Baselines.** For brevity, we will refer to the policy that utilizes the high-level policy learned with the OTA value as OTA. We compare OTA against six representative offline GCRL methods included in OGBench. Goal-conditioned behavioral cloning (**GCBC**) [11] is a simple behavior cloning method that directly imitates actions from the dataset conditioned on the goal. Goal-conditioned implicit V-learning (**GCIVL**) and goal-conditioned implicit Q-learning (**GCIQL**) [21, 33] estimate the goal-conditioned optimal value function using IQL-based expectile regression, and extract policies using AWR [36] and behavior-regularized deep deterministic policy gradient (DDPG+BC) [10], respectively. Quasimetric RL (**QRL**) [50] learns a value function that estimates the undiscounted temporal distance between state and goal via quasimetric learning and trains a policy using DDPG+BC. Contrastive RL (**CRL**) [9] approximates the Q-function via contrastive learning between state-action pairs and future states from the same trajectory, and trains the policy using DDPG+BC. **HIQL** [33] extends GCIVL with a hierarchical policy, as detailed in Section 3.

### 6.2 Evaluation on OGBench

We evaluate success rates on 14 datasets, including {`AntMaze`, `HumanoidMaze`}-{`medium`, `large`, `giant`}-{`navigate`, `stitch`} and `AntMaze`-{`medium`, `large`}-`explore`. For both `AntMaze` and `HumanoidMaze`, we report the average success rate grouped by maze size. Additionally, for visual robotic manipulation, we evaluate the average performance across four tasks: `Visual-Cube`-{`single`, `double`, `triple`} and `Visual-Scene`. As shown in Figure 4, most non-hierarchical baselines (*i.e.*, GCBC, GCIVL, GCIQL, QRL, CRL) consistently fail on long-horizon tasks. While HIQL, a hierarchical policy, achieves up to 40% success on challenging tasks such as `AntMaze-giant` and `HumanoidMaze-large`, its performance drops significantly

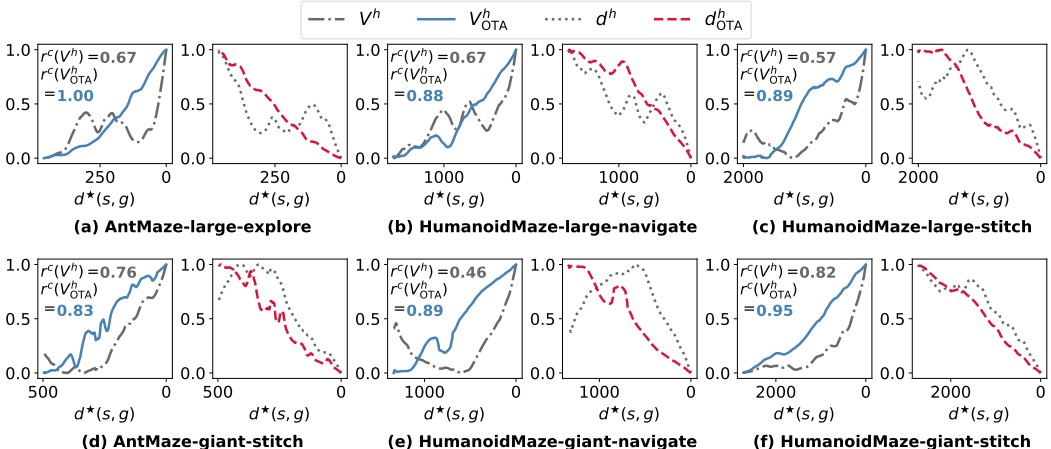

Figure 5: **Value and temporal distance estimation.** We visualize min-max normalized $V^h$, $V^h_{\text{OTA}}$, $d^h$, and $d^h_{\text{OTA}}$, and the order consistency ratios $r^c(V^h)$ and $r^c(V^h_{\text{OTA}})$, across six different datasets.

in the most difficult setting, `HumanoidMaze-giant`, highlighting its limitations in long-horizon settings.

In contrast, we observe that OTA achieves a significant performance improvement over all baselines. Notably, as the maze size increases (*i.e.*, from `medium` to `large` to `giant`), the performance gap between OTA and other methods widens substantially. These results suggest that OTA performs effective temporal abstraction and enhances high-level policy performance, even as task horizons become longer. Full benchmark results, including the `PointMaze` tasks, are provided in Appendix D.

### 6.3 High-level Value Function Visualization

In Figure 5, we compare the high-level value function $V^h$ learned with HIQL and $V^h_{\text{OTA}}$ learned with OTA across six challenging tasks. Using the visualization method from Figure 2, we plot $V^h$ and $V^h_{\text{OTA}}$ along optimal long-horizon trajectories $\tau^\star$, together with the corresponding temporal distances $d^h$ and $d^h_{\text{OTA}}$. The figure clearly shows that $V^h_{\text{OTA}}$ exhibits a more monotonic increase than $V^h$, particularly when the distance between $s$ and $g$ is large. To quantify this improvement, we compute the *order consistency ratio* $r^c$, which measures how reliably value estimates from $(s_t, s_{t+k}, g) \in \tau^\star$ produce directionally correct signals for high-level advantage estimation. Specifically, $r^c(V) = \sum_{t=0}^{T-k} \mathbf{1}\{V(s_{t+k}, g) > V(s_t, g)\}/(T - k + 1)$, where $g$ is fixed and $s_t, s_{t+k} \in \tau^\star$. Across all tasks, we observe that $r^c(V^h_{\text{OTA}}) > r^c(V^h)$, indicating that OTA yields more order-consistent value estimates.[9] Therefore, we confirm that OTA improves high-level value estimation in long-horizon tasks, leading to better high-level policy learning.

### 6.4 Effect of Varying Abstraction Factor $n$

Learning the value function $V^h_{\text{OTA}}$ depends on the abstraction factor $n$, which determines the degree of temporal abstraction. Figure 6(a-c) illustrates how the value function changes as $n$ is varied across $1, 2, 3, 5, 10,$ and $20$ in Equation 4, while keeping the optimal trajectory and goal fixed for each dataset. As shown in Figure 6(b,c), for long-horizon trajectories (*i.e.*, those exceeding a length of 1500), the absolute scale of the value function increases with larger $n$. This trend arises since the option termination condition introduces a reward of $-1$ every $n$ steps, which effectively compresses the value range as $n$ increases.

Temporal abstraction not only changes the scale of the value function but also impacts the quality of the value estimation. Figure 6(a-c) shows that when $n = 1$, the value function fails to learn as $d^\star(s, g)$ increases, which aligns with limitations commonly observed in standard HIQL. However, as $n$ increases, the value function becomes more suitable for long-horizon tasks. To further evaluate the effect of temporal abstraction, we examine the order consistency ratio $r^c$, as shown in Figure 6(d),

---

[9]We set $k = 25$ for `AntMaze` environment and $k = 100$ for `HumanoidMaze` environment.

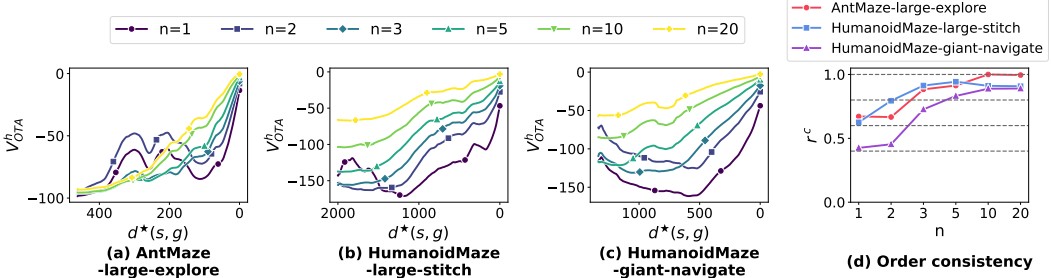

Figure 6: **Value estimation and order consistency.** (a–c) Estimation of the value function $V_{\text{OTA}}^h$ with varying abstraction factor $n$ (d) Order consistency ratio $r^c(V_{\text{OTA}}^h)$ across different values of $n$.

Table 1: **Average success rate and order consistency ratio.** Simply using $n$-step TD learning or increasing the discount factor in HIQL is insufficient to achieve the performance improvements of OTA. Here, the dataset `ALE` refers to `AntMaze-large-explore`, `AGS` to `AntMaze-giant-stitch`, `HLS` to `HumanoidMaze-large-stitch`, and `HGS` to `HumanoidMaze-giant-stitch`.

| Datasets | Success rates | | | | Order consistency ratios $r^c$ | | | |
| | HIQL | | | OTA | HIQL | | | OTA |
| | 1-step, $\gamma$ | $n$-step, $\gamma$ | 1-step, $\gamma^{1/n}$ | | 1-step, $\gamma$ | $n$-step, $\gamma$ | 1-step, $\gamma^{1/n}$ | |
|---|---|---|---|---|---|---|---|---|
| ALE | 4 $\pm5$ | 0 $\pm0$ | 3 $\pm3$ | **75** $\pm16$ | 0.75 $\pm0.01$ | 0.77 $\pm0.01$ | 0.76 $\pm0.02$ | **0.97** $\pm0.01$ |
| AGS | 2 $\pm2$ | 0 $\pm0$ | 0 $\pm0$ | **37** $\pm6$ | 0.91 $\pm0.01$ | 0.84 $\pm0.02$ | 0.79 $\pm0.02$ | **0.94** $\pm0.01$ |
| HLS | 12 $\pm4$ | 50 $\pm4$ | 22 $\pm3$ | **57** $\pm3$ | 0.76 $\pm0.01$ | 0.76 $\pm0.00$ | 0.75 $\pm0.02$ | **0.89** $\pm0.03$ |
| HGS | 28 $\pm3$ | 2 $\pm2$ | 2 $\pm1$ | **79** $\pm3$ | 0.71 $\pm0.01$ | 0.72 $\pm0.00$ | 0.72 $\pm0.01$ | **0.94** $\pm0.01$ |

which generally increases with $n$. However, beyond a certain point, larger $n$ causes a drop in $r^c(V^h)$, indicating that excessive temporal abstraction may lead to a loss of information.

## 6.5 Impact of $n$-Step TD and Increasing the Discount Factor $\gamma$

In the original HIQL, the high-level value function $V^h$ is discounted by $\gamma$ at every step. In contrast, the OTA value function $V_{\text{OTA}}^h$ applies discounting only every $n$ steps. To investigate the source of the effectiveness of OTA, we modify the value learning approach of standard HIQL in two ways: (1) using $n$-step TD learning, and (2) increasing $\gamma$. We evaluate both the success rate and the order consistency ratio $r^c$ across four datasets. In Table 1, we set $n = 15$ for `AntMaze` and $n = 20$ for `HumanoidMaze`. To compute $r^c$, we collect 5 trajectories per dataset and report the average consistency ratio (see Appendix B.2.3 for details of the collected trajectories).

The first variant uses the original $\gamma$ with $n$-step TD learning in HIQL, denoted as HIQL($n$-step, $\gamma$). Table 1 shows that this approach yields almost no improvement in $r^c$, and the success rates also show little gain except for `HumanoidMaze-large-stitch`. These results indicate that $n$-step TD targets still suffer from value function estimation errors when the discount factor remains unchanged.

The second variant keeps 1-step TD learning but modifies the discount factor to $\gamma^{1/n}$. Under OTA training, the optimal value function becomes $V^\star(s_t, g) = -(1 - \gamma^{d^\star(s_t,g)/n})/(1 - \gamma)$. Therefore, to approximate this temporally abstracted optimal value function, a possible approach is to increase the discount factor $\gamma$ to $\gamma^{1/n}$ in Equation (1). However, Table 1 shows that simply increasing $\gamma$ fails to outperform standard HIQL in either success rate or $r^c$. In contrast, OTA achieves significant gains in long-horizon tasks such as `HumanoidMaze-giant-stitch`. The experiments demonstrate that simply adjusting the discounting factor alone is insufficient, and the temporal abstraction is crucial for effective value learning in long-horizon tasks.

Our analysis further suggests that $n$-step TD learning could potentially be improved by carefully adjusting $\gamma$ for each $n$. However, this would introduce additional complexity in hyperparameter selection. In contrast, OTA fixes $\gamma$ regardless of $n$, which makes the approach much simpler.

## 6.6 Scalability Comparison of TD-Based OTA and QRL

Here, we demonstrate that OTA, which leverages a TD-based IQL loss, scales effectively with increasing state and action dimensionality. As discussed in Section 4.2, conventional TD methods rely on a discount factor, which causes the advantage estimate to decay exponentially over long horizons. To avoid this issue, we explore alternative value learning approaches that do not depend on a discount factor.

In particular, we consider QRL, which learns *undiscounted* temporal distances between states through quasimetric learning (see Appendix C for more details). However, QRL relies on min-max optimization, which becomes computationally challenging in high-dimensional state spaces. As shown in

Table 2: **Success rates for different high-level values.**

| Datasets | QRL | HIQL | OTA |
|---|---|---|---|
| AntMaze-giant-navigate | 76 ±2 | 65 ±5 | **77** ±4 |
| HumanoidMaze-giant-navigate | 12 ±3 | 12 ±4 | **92** ±0 |
| Visual-cube-double | 6 ±2 | 59 ±3 | **65** ±2 |
| Visual-scene | 5 ±2 | 50 ±1 | **54** ±2 |

Table 2, QRL achieves significantly lower success rates on complex tasks such as `HumanoidMaze` and `Visual-scene`. These results highlight the scalability and the practical advantages of our TD-based OTA, particularly in environments with high-dimensional state spaces.

## 7 Conclusion

In this paper, we investigated the limitations of the hierarchical offline GCRL method HIQL, particularly in long-horizon tasks. Our analysis revealed that the main performance bottleneck lies in the high-level policy, which suffers from inaccurate value estimates when the state-goal distance is large. To address this challenge, we proposed OTA, a method that incorporates temporal abstraction into IQL-based value learning through the concept of options. Experiments on challenging long-horizon goal-reaching tasks show that OTA enables high-level policies to achieve substantial performance improvements in long-term planning. The simplicity and effectiveness of OTA highlight its potential for advancing long-horizon offline GCRL in real-world applications.

## Acknowledgments

This work was supported in part by National Research Foundation of Korea (NRF) grant [No. 2021R1A2C2007884, No. RS-2025-02263628], the Institute of Information & communications Technology Planning & Evaluation (IITP) grants [RS-2021-II212068, RS-2022-II220113, RS-2022-II220959, RS-2021-II211343], and the BK21 FOUR Education and Research Program for Future ICT Pioneers (Seoul National University), funded by the Korean government (MSIT).

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

Figure 7: **Dataset examples.** For `Maze` environment, the task differ by (a) environment type (b) and dataset type. (c) In `Visual-cube`, the robot must manipulate the cube to the location specified by the blurred cube, which denotes the goal position.

## A   Limitations

Our method, OTA, has several following limitations. First, we introduce a new hyperparameter, temporal abstraction factor $n$, to reduce the effective horizon of the value function. Due to the additional hyperparameter, we should carefully select both $k$, the number of steps to reach subgoal, and $n$. Second, though we carry out temporal abstraction on the value function, we still cannot obtain an order consistent value function for all state and goal pairs. Third, for the experiments on long-horizon tasks in which the trajectory length is more than 1000, we only use the maze dataset to evaluate our method.

## B   Experimental Details

### B.1   Environments, Tasks, and Datasets

In this section, we provide detailed descriptions of each task, with dataset examples illustrated in Figure 7. For a more detailed description of the environment, see OGBench [32].

**Maze** (`Maze`) is a challenging long-horizon locomotion task, where the agent needs to reach the given goal position from the initial position. This environment is categorized into three different types of agent based on state and action dimension: 1) Pointmaze (`PointMaze`), which controls 2 degrees of freedom (DoF) point mass, 2) Antmaze (`AntMaze`), which controls a quadrupedal Ant with 8-DoF, and 3) Humanoidmaze (`HumanoidMaze`), which controls 21-DoF Humanoid agent. Each maze environment is divided into medium, large, and giant based on maze size, from `PointMaze-medium` requiring a horizon length (*i.e.*, maximum episode steps) of 1000, to `HumanoidMaze-giant` requiring 4000. Each environment includes diverse datasets—`navigate`, `stitch`, and `explore`—collected via different dataset features:

- `navigate`: This dataset consists of trajectories collected as an agent, guided by a noisy expert policy, that attempted to reach randomly sampled goals.

Table 3: **Common hyperparameters for OTA.** We refer to Appendix B.2.1 hyperparameter definition.

| Hyperparameter | Value |
|---|---|
| Learning rate | 3e-4 |
| Optimizer | Adam |
| Minibatch size | 1024 (`Maze`), 256 (`Visual env`) |
| Total gradient steps | 1000000 (`Maze`), 500000 (`Visual env`) |
| MLP dimensions | [512, 512, 512] |
| Activation function | GELU |
| Target network smoothing coefficient | 0.005 |
| Discount factor $\gamma$ | 0.99 (default), 0.995 (`Antmaze-giant`, `HumanoidMaze`) |
| Image augmentation probability | 0.5 (random crop) |
| Policy ($p_{\text{cur}}^{\mathcal{D}}, p_{\text{traj}}^{\mathcal{D}}, p_{\text{rand}}^{\mathcal{D}}$) ratio | (0,1,0) (default), (0,0.5, 0.5) (`stitch`), (0,0,1) (`explore`) |
| Value ($p_{\text{cur}}^{\mathcal{D}}, p_{\text{traj}}^{\mathcal{D}}, p_{\text{rand}}^{\mathcal{D}}$) ratio | (0.2, 0.5, 0.3) |

- `stitch`: This dataset contains shorter trajectories compared to those collected in the `navigate` setting. They are designed to evaluate goal-stitching capabilities.
- `explore`: This includes higher levels of action noise, resulting in lower-quality data, but with increased state coverage.

**Visual-cube** (`Visual-cube`) is a challenging robotic visual manipulation task, where the agent must move and stack cube blocks to reach a specified goal configuration. The task includes three variants—`single`, `double`, and `triple`—corresponding to the number of cubes that must be manipulated. The agent receives pixel-based images of the current observation and goal, each of size $64 \times 64 \times 3$, and outputs a 5-DoF action vector. The task horizon ranges from 200 steps (`single`) to 1000 steps (`triple`). The agent is learned with noisy dataset, which is built from a suboptimal dataset with action noise, leading to extremely low-quality data and longer effective horizons.

**Visual-scene** (`Visual-scene`) is also a robotic visual manipulation task, where the agent needs to manipulate everyday objects -a window, a drawer, two button locks—where pressing a button toggles the lock status of the corresponding object (*i.e.*, the drawer or the window). The agent receives pixel-based images of the current observation and goal, each of size $64 \times 64 \times 3$, and outputs a 5-DoF action vector. The task horizon range is 750, in that it involves unlocking object and manipulating the object. The agent is learned with noisy dataset, as mentioned above.

## B.2 Implementation Details

### B.2.1 Hyperparameters

We implemented OTA on top of the official implementation of OGBench [32][10]. We use goal-sampling distribution for value and policy learning, following OGBench. Data sampling scheme is based on HER [1], taking three different goal-sampling distributions, definition is as follows:

- $p_{\text{cur}}^{\mathcal{D}}(g|s)$ is a Dirac delta distribution centered at the current state $s$ (*i.e.*, $g = s$),
- $p_{\text{traj}}^{\mathcal{D}}(g|s)$ is the probability distribution over goals $g$, where each goal is uniformly sampled from the future states within the same trajectory as the state $s$,
- $p_{\text{rand}}^{\mathcal{D}}(g|s)$ is the probability distribution that a goal $g$ is uniformly sampled from the entire dataset $\mathcal{D}$.

Task-specific hyperparameters are organized in Table 4, where hyperparameters are described in Equation (1) to Equation (4).

---

[10]https://github.com/seohongpark/ogbench

| Task category | | | OTA hyperparameters | | | |
|---|---|---|---|---|---|---|
| Environment | Type | Size | $\beta^h$ | $\beta^\ell$ | $k$ | $n$ |
| **Maze** | | | | | | |
| PointMaze | navigate | medium | 0.5 | 3.0 | 25 | 5 |
| | | large | 3.0 | 3.0 | 25 | 5 |
| | | giant | 3.0 | 3.0 | 20 | 5 |
| | stitch | medium | 1.0 | 3.0 | 20 | 4 |
| | | large | 1.0 | 3.0 | 20 | 10 |
| | | giant | 5.0 | 3.0 | 20 | 5 |
| AntMaze | navigate | medium | 1.0 | 3.0 | 25 | 5 |
| | | large | 1.0 | 3.0 | 25 | 5 |
| | | giant | 0.5 | 3.0 | 16 | 4 |
| | stitch | medium | 0.5 | 3.0 | 25 | 5 |
| | | large | 1.0 | 3.0 | 25 | 5 |
| | | giant | 3.0 | 3.0 | 30 | 10 |
| | explore | medium | 3.0 | 3.0 | 25 | 5 |
| | | large | 3.0 | 3.0 | 20 | 15 |
| HumanoidMaze | navigate | medium | 0.5 | 3.0 | 100 | 20 |
| | | large | 0.5 | 3.0 | 100 | 20 |
| | | giant | 0.5 | 3.0 | 100 | 20 |
| | stitch | medium | 3.0 | 3.0 | 100 | 20 |
| | | large | 1.0 | 3.0 | 100 | 20 |
| | | giant | 0.5 | 3.0 | 100 | 20 |
| **Robotic visual manipulation** | | | | | | |
| Visual-cube | noisy | single | 1.0 | 3.0 | 20 | 4 |
| | | double | 3.0 | 3.0 | 20 | 4 |
| | | triple | 3.0 | 3.0 | 25 | 4 |
| Visual-scene | noisy | | 3.0 | 3.0 | 10 | 4 |

Table 4: **Task specific hyperparameters for OTA.** We refer to Appendix B.2.1 for each hyperparameter variable. Note that we individually tune the hyperparameters for each task.

| Task category | | Maximum episode length |
|---|---|---|
| Environment | Size | |
| **Maze** | | |
| PointMaze | medium | 1000 |
| | large | 1000 |
| | giant | 1000 |
| AntMaze | medium | 1000 |
| | large | 1000 |
| | giant | 1000 |
| HumanoidMaze | medium | 2000 |
| | large | 2000 |
| | giant | 4000 |
| **Robotic visual manipulation** | | |
| Visual-cube | single | 200 |
| | double | 500 |
| | triple | 1000 |
| Visual-scene | | 750 |

Table 5: **Maximum episode length of environments.**

### B.2.2 Training and Evaluation Details

In `Maze` environment, the model is trained for up to 1M gradient steps. We evaluate the model at 800K, 900K, and 1M steps. At each evaluation point, we measure the success rate using five fixed task goals provided by OGBench. Each goal is evaluated with 50 rollouts, resulting in 750 evaluation episodes per seed (*i.e.*, 3 evaluation steps $\times$ 5 goals $\times$ 50 rollouts). We report the average success rate across these episodes and across 8 different random seeds. For `Visual-cube` and `Visual-scene` environments, the model is trained for 500K gradient steps. Evaluations are conducted at 300K, 400K, and 500K steps using the same protocol: five fixed goals and 50 rollouts per goal. The maximum episode length of each environment is shown in the Table 5. All results are averaged across 8 seeds. All experiments are conducted using NVIDIA RTX A5000 and A6000 GPUs.

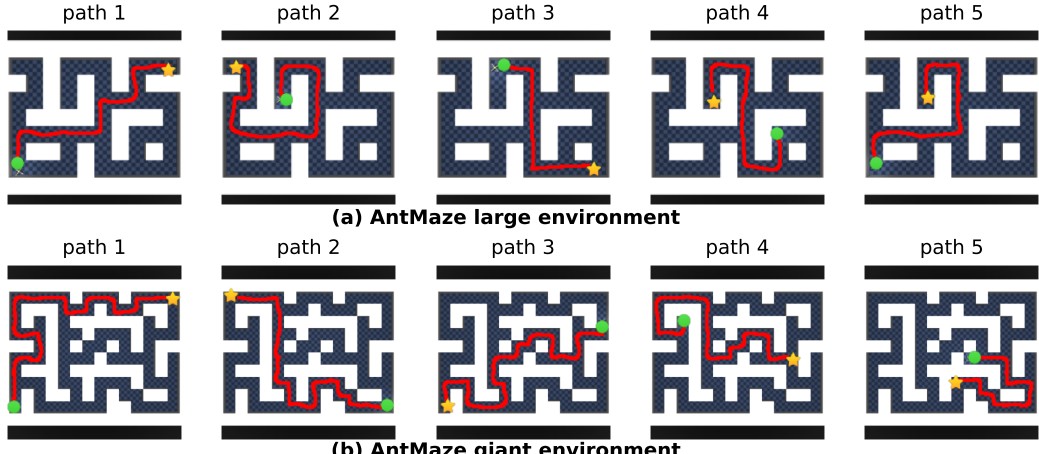

**(a) AntMaze large environment**

**(b) AntMaze giant environment**

Figure 8: **Collected optimal trajectories for AntMaze environment.** We collect the optimal trajectories from the initial state (🟢) to the goal (⭐)

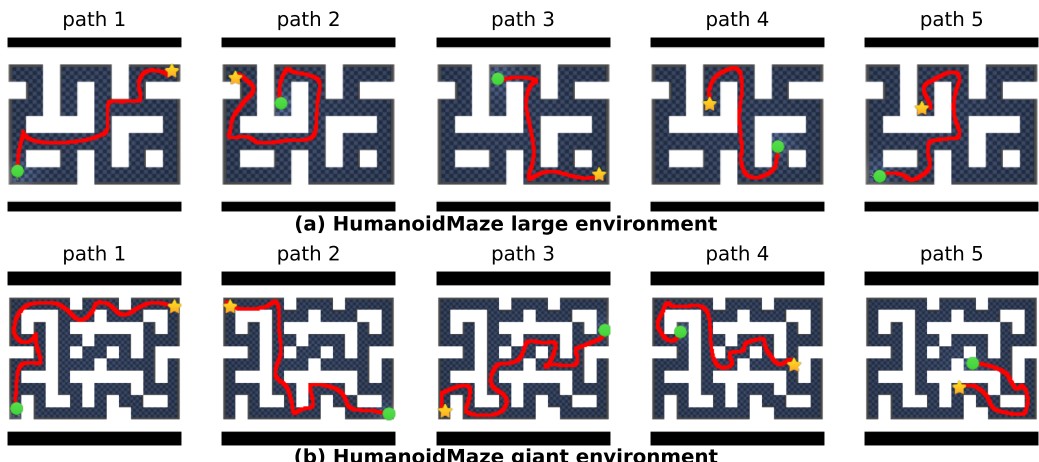

**(a) HumanoidMaze large environment**

**(b) HumanoidMaze giant environment**

Figure 9: **Collected optimal trajectories for HumanoidMaze environment.** We collect the optimal trajectories from the initial state (🟢) to the goal (⭐)

### B.2.3   Collected Optimal Trajectories

To evaluate the order consistency of value for high-level advantage, we collect five optimal trajectories for each environment: `AntMaze-{large, giant}` and `HumanoidMaze-{large, giant}`. Each optimal trajectory is generated using the expert policy that was originally used during the offline dataset collection in OGBench.

The collected optimal trajectories for `AntMaze` and `HumanoidMaze` are shown in Figures 8 and 9, respectively. Order consistency, as reported in Table 1, is evaluated based on the five trajectories illustrated in these figures and averaged over 8 random seeds. During value estimation, we apply a moving average with an appropriate temporal window size to smooth out short-term fluctuations and obtain stable value estimates. The optimal trajectories used for value visualizations in Figures 5 and 6 are as follows:

**Trajectory selection for Figure 5:**

- Figure 5(a): path 5
- Figure 5(b): path 5
- Figure 5(c): path 2
- Figure 5(d): path 5
- Figure 5(e): path 5

- Figure 5(f): path 1

**Trajectory selection for Figure 6:**

- Figure 6(a): path 5
- Figure 6(b): path 2
- Figure 6(c): path 5

## C    Quasimetric Reinforcement Learning (QRL)

QRL [50] is an goal-conditioned RL algorithm by utilizing the quasimetric structure for learning optimal value function $V^\star$. The quasimetrics are a generalization of metrics in that they do require symmetry. The optimal value function in QRL is an *undiscounted* temporal distance, $V^\star(s, g) = -d^\star(s, g)$, and the value function satisfies the triangular inequality, $d^\star(s, s') + d^\star(s', g) \geq d^\star(s, g)$ for any $s, s' \in \mathcal{S}$, and $g \in \mathcal{G}$. To obtain the optimal value function using the quasimetric structure, the value function should have two properties: First, the value function should should have **locally consistent value**, $d^\star(s, s') \leq -r$. Second, **the distance should be globally spread out**, $d^\star(s, g) = $ *total cost of path connecting s to g*. To achieve those properties, QRL optimizes the following objective to obtain the optimal value function:

$$\min_{\theta} \max_{\lambda \geq 0} -\mathbb{E}_{(s,g)\sim\mathcal{D}}[\phi(d_\theta^{\text{IQE}}(s, g))] + \lambda\big(\mathbb{E}_{(s,a,s',r)\sim\mathcal{D}}[\text{relu}(d_\theta^{\text{IQE}}(s, s') + r)^2] - \epsilon^2\big), \quad (5)$$

where $\phi$ is a monotonically increasing convex function, $d^{\text{IQE}}(\cdot, \cdot)$ is Interval Quasimetric Embeddings (IQE) [49] for the quasimetric model. In the above objective, both the min and max operations should be applied simultaneously, which can induce unstable training. Using the value function, QRL learns policy through optimizing the DDPG + BC [10] like objective.

# D  Additional Results

## D.1  Per-environment Results

We show the full per-environment results in Table 6. In this table, OTA outperforms the baselines in most cases.

| Task category | | | Non-hierarchical | | | | | Hierarchical | |
|---|---|---|---|---|---|---|---|---|---|
| Environment | Type | Size | GCBC | GCIVL | GCIQL | QRL | CRL | HIQL | OTA |
| **Maze** | | | | | | | | | |
| PointMaze | navigate | medium | 9 ±6 | 63 ±6 | 53 ±8 | 82 ±5 | 29 ±7 | 79 ±5 | **86** ±2 |
| | | large | 29 ±6 | 45 ±5 | 34 ±3 | **86** ±9 | 39 ±7 | 58 ±5 | 85 ±5 |
| | | giant | 1 ±2 | 0 ±0 | 0 ±0 | 68 ±7 | 27 ±10 | 46 ±9 | **72** ±6 |
| | stitch | medium | 23 ±18 | 70 ±14 | 21 ±9 | 80 ±12 | 0 ±1 | 74 ±6 | **75** ±5 |
| | | large | 7 ±5 | 12 ±6 | 31 ±2 | 84 ±15 | 0 ±0 | 13 ±6 | **66** ±8 |
| | | giant | 0 ±0 | 0 ±0 | 0 ±0 | 50 ±8 | 0 ±0 | 0 ±0 | **52** ±7 |
| AntMaze | navigate | medium | 29 ±4 | 72 ±8 | 71 ±4 | 88 ±3 | 95 ±1 | **96** ±1 | **96** ±1 |
| | | large | 24 ±2 | 16 ±5 | 34 ±4 | 75 ±6 | 83 ±4 | 91 ±2 | **92** ±1 |
| | | giant | 0 ±0 | 0 ±0 | 0 ±0 | 14 ±3 | 16 ±3 | 65 ±5 | **77** ±4 |
| | stitch | medium | 45 ±11 | 44 ±6 | 29 ±6 | 59 ±7 | 53 ±6 | **94** ±1 | 93 ±1 |
| | | large | 3 ±3 | 18 ±2 | 7 ±2 | 18 ±2 | 11 ±2 | 67 ±5 | **84** ±3 |
| | | giant | 0 ±0 | 0 ±0 | 0 ±0 | 0 ±0 | 0 ±0 | 2 ±2 | **37** ±6 |
| | explore | medium | 2 ±1 | 19 ±3 | 13 ±2 | 1 ±1 | 3 ±2 | 37 ±10 | **94** ±3 |
| | | large | 0 ±0 | 10 ±3 | 0 ±0 | 0 ±0 | 0 ±0 | 4 ±5 | **75** ±16 |
| HumanoidMaze | navigate | medium | 8 ±2 | 24 ±2 | 27 ±2 | 21 ±8 | 60 ±4 | 89 ±2 | **94** ±1 |
| | | large | 1 ±0 | 2 ±1 | 2 ±1 | 5 ±1 | 24 ±4 | 49 ±4 | **83** ±2 |
| | | giant | 0 ±0 | 0 ±0 | 0 ±0 | 1 ±0 | 3 ±2 | 12 ±4 | **92** ±1 |
| | stitch | medium | 29 ±5 | 12 ±2 | 12 ±3 | 18 ±2 | 36 ±2 | 88 ±2 | **88** ±2 |
| | | large | 6 ±3 | 1 ±1 | 0 ±0 | 3 ±1 | 4 ±1 | 28 ±3 | **57** ±3 |
| | | giant | 0 ±0 | 0 ±0 | 0 ±0 | 0 ±0 | 0 ±0 | 3 ±2 | **79** ±3 |
| **Robotic visual manipulation** | | | | | | | | | |
| Visual-cube | noisy | single | 14 ±3 | 75 ±3 | 48 ±3 | 10 ±5 | 39 ±30 | 99 ±0 | **99** ±0 |
| | | double | 5 ±1 | 17 ±4 | 22 ±2 | 6 ±2 | 6 ±3 | 59 ±3 | **65** ±2 |
| | | triple | 16 ±1 | 18 ±1 | 12 ±1 | 9 ±4 | 16 ±1 | 23 ±2 | **26** ±2 |
| Visual-scene | noisy | | 13 ±2 | 23 ±2 | 12 ±4 | 2 ±0 | 15 ±2 | 50 ±1 | **54** ±2 |

Table 6: **Performance comparison across various policy types and benchmarks.** We shot average success rate on 8 random seeds. Bold values indicate the best performance in each row. Baseline performances are the official results provided by OGBench.

## D.2 Performance under Unified Hyperparameters

We report additional results where OTA is trained with the same hyperparameters as HIQL, except for the temporal abstraction factor $n$. The experiments are conducted on complex maze environments.

As shown in Table 7, OTA consistently outperforms HIQL even under identical hyperparameter settings. This result indicates that incorporating temporal abstraction alone significantly enhances performance in long-horizon goal-conditioned tasks.

| Task category | | | Hyperparameters | | | | Methods | |
|---|---|---|---|---|---|---|---|---|
| Environment | Type | Size | $n$ | $k$ | $\beta^h$ | $\beta^\ell$ | HIQL | OTA |
| PointMaze | navigate | large | 5 | 25 | 3.0 | 3.0 | $58\pm5$ | $\mathbf{85}\pm5$ |
| | | giant | 5 | 25 | 3.0 | 3.0 | $46\pm9$ | $\mathbf{72}\pm6$ |
| | stitch | large | 5 | 25 | 3.0 | 3.0 | $13\pm6$ | $\mathbf{46}\pm7$ |
| | | giant | 5 | 25 | 3.0 | 3.0 | $0\pm0$ | $\mathbf{44}\pm8$ |
| AntMaze | navigate | large | 5 | 25 | 3.0 | 3.0 | $91\pm2$ | $\mathbf{91}\pm1$ |
| | | giant | 5 | 25 | 3.0 | 3.0 | $65\pm5$ | $\mathbf{70}\pm2$ |
| | stitch | large | 5 | 25 | 3.0 | 3.0 | $67\pm5$ | $\mathbf{79}\pm3$ |
| | | giant | 5 | 25 | 3.0 | 3.0 | $2\pm2$ | $\mathbf{29}\pm5$ |
| | explore | medium | 5 | 25 | 3.0 | 3.0 | $37\pm10$ | $\mathbf{93}\pm3$ |
| | | large | 10 | 25 | 3.0 | 3.0 | $4\pm5$ | $\mathbf{62}\pm12$ |
| HumanoidMaze | navigate | large | 20 | 100 | 3.0 | 3.0 | $49\pm4$ | $\mathbf{82}\pm2$ |
| | | giant | 20 | 100 | 3.0 | 3.0 | $12\pm4$ | $\mathbf{91}\pm1$ |
| | stitch | large | 20 | 100 | 3.0 | 3.0 | $28\pm3$ | $\mathbf{43}\pm3$ |
| | | giant | 20 | 100 | 3.0 | 3.0 | $3\pm2$ | $\mathbf{61}\pm3$ |

Table 7: **Performance under unified hyperparameters.**

## D.3 Performance on Visual Play Datasets

We evaluate the performance of OTA on visual play datasets, with results summarized in Table 8. For a fair comparison, we fix the hyperparameters $(k, \beta^h, \beta^l) = (10, 3.0, 3.0)$ for both HIQL and OTA, varying only $n = 2$.

| Task category | | | Methods | |
|---|---|---|---|---|
| Environment | Data | Env | HIQL | OTA |
| Visual-cube | play | double | $48\pm4$ | $\mathbf{51}\pm3$ |
| | play | triple | $21\pm5$ | $\mathbf{28}\pm1$ |
| Visual-scene | play | - | $50\pm5$ | $\mathbf{56}\pm5$ |

Table 8: **Performance comparison on visual manipulation play dataset.**

