# OpenReview forum: "Option-aware Temporally Abstracted Value for Offline Goal-Conditioned Reinforcement Learning"
_NeurIPS.cc/2025/Conference — NeurIPS 2025 spotlight_

### Official Review · Reviewer_qrJ2 · 2025-06-01

**Clarity:** 3
**Significance:** 3
**Originality:** 3
**Rating:** 5
**Confidence:** 4

**Summary:**

This paper investigates the performance bottleneck of the prior method HIQL and concludes that inaccurate value estimation causes the high-level policy to suggest suboptimal subgoals. Further analysis reveals that this inaccuracy arises from the large temporal distance between the current state and the goal. To address this, the authors propose a novel, simple, and effective method called OTA, which estimates the value function over a sequence of actions, effectively transforming distant goals into nearer ones. Experiments show that this technique improves order consistency, meaning that states can be ranked by the value function in the same order as their temporal sequence, and outperforms the existing methods.

**Questions:**

1. In the abstract, the authors state that offline GCRL trains goal-reaching policies from unlabeled (reward-free) datasets. However, the datasets used in offline GCRL include binary rewards / reward function, that indicates whether the goal has been reached. Therefore, describing them as "reward-free" may be inaccurate and should be clarified.

2. The term "policy extraction" is used multiple times (e.g., line 29 and line 149) without a prior explanation. It would improve clarity if the authors could briefly define what they mean by policy extraction in this context.

3. The reward function defined in lines 121 and 207 as $r(s, g) = -1(s \neq g)$ assumes that exact equality $s = g$ is achievable. This assumption is problematic in continuous state spaces, where reaching the exact goal is often impossible. A more realistic formulation would be $r(s, g) = -1(\|s - g\|^2 > \epsilon)$, for some threshold $\epsilon > 0$.

4. In Figure 2, it appears that when the state $s$ is far from the goal $g$, the high-level value function $V^h$ tends to yield over-optimistic estimates. This behavior is noteworthy, and it would strengthen the paper if the authors could provide an explanation or hypothesis for why this phenomenon occurs.

**Ethical Concerns:**

["NO or VERY MINOR ethics concerns only"]

**Final Justification:**

The authors have addressed my concerns and we had a nice discussion.

**Limitations:**

Yes

**Quality:**

4

**Strengths And Weaknesses:**

**Strength**

1. Section 4 is compelling and well-argued, offering a strong and intuitive motivation for the proposed method.

2. The authors introduce a new concept, "order consistency", which is both interesting and potentially impactful for future research in goal-conditioned reinforcement learning.

3. The experimental results are thorough and convincingly demonstrate the effectiveness of the proposed approach.

**Weakness**

The paper does not provide sufficient theoretical justification for the OTA value function $V^h_{\text{OTA}}$ as optimised in Eq. (4). Please try to address the following questions:

(1) Can the value function be interpreted as the expected sum of rewards? If not, how to define this value function?

(2) Does this formulation satisfy the Bellman recursion? It is the ground for optimisation following Eq. (4).

I would like to increase my rating to Accept once the authors address the two questions.

---

> ### Author Rebuttal · Authors · 2025-07-31
>
> We sincerely thank Reviewer qrJ2 for the positive comments on the motivation of the proposed method and “order consistency” evaluation metric. We also appreciate your constructive feedback regarding theoretical justifications, terminology, reward assumptions, etc. Below, we address your concerns in detail.
>
> ### Weakness 1: Theoretical justification for the OTA value
>
> We thank the reviewer for these important questions.
> (1) The value function in our formulation can be interpreted as the expected sum of discounted rewards—i.e., the return—which aligns with the standard definition in reinforcement learning.
> (2)  Our approach builds on the options framework, where it has been thoroughly studied that the value function converges to the optimal value under Bellman recursion. For a more detailed theoretical proof, we kindly refer the reviewer to foundational work [1].
>
> ### Question 1: Clarification of “reward-free dataset”
>
> We agree that the term *"reward-free"* may be misleading without clarification. In our setting, the offline dataset used for GCRL training consists of trajectories in the form of $(s_0, a_0, s_1, a_1, \dots)$, and does not include explicit reward signals. However, during training, we apply a sparse binary reward function using hindsight experience replay (HER) [2]. Thus, while the original dataset itself is unlabeled in terms of rewards, rewards are introduced during training via relabeling. To avoid confusion, we will replace the term "reward-free dataset" with "reward-unlabeled dataset”.
>
> ### Question 2: Clarification of “policy extraction”
>
> We use the term *“policy extraction”* to refer to the process of deriving a policy from a learned value function, a usage consistent with prior works such as IQL [3], OGBench [4], and JaxGCRL [5]. This term emphasizes the decoupling between value learning and policy learning, particularly in algorithms like AWR, which IQL builds upon.
> Specifically, we will make the following revisions:
> - Line 29 will be removed for clarity and conciseness.
> - In earlier sections (e.g., lines 49, 52, 60, ...), where we refer to general policy learning, we will consistently use *policy learning* instead of *policy extraction*.
> - Beginning with the related work section (e.g., line 123), where AWR is discussed explicitly, we will adopt the term *policy extraction* and provide a brief definition for clarity.
>
> ### Question 3: Reward function formulation
>
> We agree that in continuous state spaces, a more appropriate reward formulation would be  $r(s,g)=-\mathbf{1}(|s-g|^2>\epsilon)$. However, in practice, this formulation introduces certain challenges: (1) it introduces an additional hyperparameter $\epsilon$, (2) it increases the computational overhead in HER-style goal relabeling, especially for high-dimensional states. For these reasons, we chose to use a sparse binary reward of $-\mathbf{1}(s\neq g)$ in our experiments following other GCRL benchmarks [4,5]. We will clarify this design choice in the experimental section of the revised manuscript.
>
> ### Question 4: Over-optimistic value estimates in Figure 2
>
> Thank you for your insightful observation regarding over-optimistic estimates of the high-level value function when $s$ is far from the goal $g$. We hypothesize that this behavior arises from the value learning mechanism in IQL, which employs expectile regression to estimate the upper expectile of TD targets. As the distance between $s$ and $g$ increases, the variance (noise) of the TD targets also tends to increase (Figure 8 in HIQL [6]). Consequently, the upper expectile can become significantly biased, leading to overestimated values for distant goals. That is, the estimation error tends to accumulate over longer horizons in IQL, resulting in increasingly over-optimistic value estimates. In contrast, our OTA value learning approach effectively mitigates this issue, even as the horizon increases. We will incorporate this explanation in the revised manuscript and supplement it with additional visualizations comparing OTA with the ground-truth value function to support the claim.
>
> Reference
>
> [1] Sutton et al., "Between MDPs and semi-MDPs: A framework for temporal abstraction in reinforcement learning." Artificial intelligence 1999
>
> [2] Andrychowicz et al., “Hindsight Experience Replay”, NeurIPS 2017
>
> [3] Kostrikov et al., “Offline Reinforcement Learning with Implicit Q-Learning”, ICLR 2022
>
> [4] Park et al., “Ogbench: Benchmarking Offline Goal-Conditioned RL”, ICLR 2025
>
> [5] Bortkiewicz et al., “Accelerating Goal-Conditioned RL Algorithms and Research”, ICLR 2025
>
> [6] Park et al., “HIQL: Offline Goal-Conditioned RL with Latent States as Actions”, NeurIPS 2023

---

> ### Comment · Reviewer_qrJ2 · 2025-08-01
>
> Thanks for your replies. I have the following comments:
>
> **Weakness.** I would suggest that the authors clarify this point and cite the relevant work in the section.
>
> **Q1.** It would be better to remove *reward-free* or *reward-unlabelled*. Since the reward function is provided in offline GCRL, it is misleading to describe it as reward-free or reward-unlabelled.
>
> **Q3.** I don't think the two challenges are valid. (1) It does not introduce an additional hyperparameter — even if you don’t use my suggested definition, the threshold $\epsilon$ still appears in your code. (2) For the same reason — even without using my definition, you still need to compute the norm to obtain the reward after goal relabelling.
>
> Although these are minor issues, the authors should address them seriously.

---

> > ### Author Response · Authors · 2025-08-03
> >
> > Thank you for the constructive feedback. Guided by your comments, we will clarify our terminology and setup through the following revisions.
> >
> > 1. **Remove the term *reward-free* dataset**
> >
> > We will remove the terms *reward-free* and *reward-unlabelled* from the manuscript.
> >
> > While such terms appear in some recent works [1,2,3,4] to describe datasets without task-specific reward labels, we acknowledge that (1) this usage is still limited to a relatively small set of recent papers, and (2) it can be misleading since a reward function does exist. To avoid confusion, we will remove these terms from our manuscript.
> >
> >
> > 2. **Specify reward function formulation**
> >
> > The original HER paper discusses a norm-based criterion for goal relabelling. However, subsequent works using HER-style goal relabeling typically adopt one of two reward function formulations: either 1) a norm-based reward with an epsilon threshold [5,6,7] or 2) an exact-match reward defined as $-1(s \neq g)$ [1,2,8]. We confirmed each definition both in the papers and their official code.
> >
> > Our implementation is based on OGBench [1], which uses the exact-match reward $-1(s \neq g)$ with no epsilon threshold. Therefore, to avoid ambiguity, we will state in Preliminaries section that we use a (-1, 0)-sparse reward, and specify in Experiments section that our implementation follows the OGBench.
> >
> > > Reference
> >
> > [1] Park et al., “Ogbench: Benchmarking Offline Goal-Conditioned RL”, ICLR 2025
> >
> > [2] Wu et al., "Planning, Fast and Slow: Online Reinforcement Learning with Action-Free Offline Data via Multiscale Planners", ICML 2024
> >
> > [3] Park et al., "Foundation Policies with Hilbert Representations", ICML 2024
> >
> > [4] Yu et al., "How to Leverage Unlabeled Data in Offline Reinforcement Learning", ICML 2022
> >
> > [5] Chane-Sane et al., "Goal-Conditioned Reinforcement Learning with Imagined Subgoals", ICML 2021
> >
> > [6] Ma et al., "Offline Goal-Conditioned Reinforcement Learning via f-Advantage Regression", NeurIPS 2022
> >
> > [7] Jain et al., "Learning to Reach Goals via Diffusion", ICML 2024
> >
> > [8] Chebotar et al., "Actionable Models: Unsupervised Offline Reinforcement Learning of Robotic Skills", ICML 2021

---

> > > ### Comment · Reviewer_qrJ2 · 2025-08-03
> > >
> > > Thanks for your response. Are you certain that [1, 2, 8] define the exact-match reward both in the papers and in their official code? And are you sure that OGBench's implementation on which you based uses exact-match without any epsilon threshold? I would suggest you review your code carefully.
> > >
> > > Could you please check the latest version of OGBench (v1.1.5, released on 3 July)? It includes a file called `relabel_utils.py`, which defines the dataset relabelling function `relabel_dataset`. Based on lines 25, 27, and 50, I don’t believe your claim is accurate.
> > >
> > > Additionally, while I understand that some papers define the reward function as exact-match, that doesn’t necessarily imply it is the most appropriate or accurate choice.

---

> > > > ### Author Response · Authors · 2025-08-03
> > > >
> > > > Thank you for your response.
> > > >
> > > > The ``relabel_utils.py`` file is designed for single-task (non-goal-conditioned) offline RL provided by OGBench. It is used to compute rewards for offline RL from the collected trajectory dataset.
> > > >
> > > > The file related to computing rewards suitable for offline GCRL, which matches our experimental setting, is ``impls/utils/datasets.py``. As shown in lines 351–353, the reward is computed based on exact-match criterion.
> > > >
> > > > While we followed the official code, we agree that a norm-based reward is a more accurate choice. Therefore, we will include an experiment in the revised manuscript to investigate whether changing the reward leads to any performance differences.

---

> > > > > ### Comment · Reviewer_qrJ2 · 2025-08-03
> > > > >
> > > > > Thanks for your responses. I've checked the code. The line 351-353 show
> > > > >
> > > > > ```351: successes = (idxs == value_goal_idxs).astype(float)```
> > > > >
> > > > > ```352: batch['masks'] = 1.0 - successes)```
> > > > >
> > > > > ```353: batch['rewards'] = successes - (1.0 if self.config['gc_negative'] else 0.0)```
> > > > >
> > > > > Here the exact-match is on the index --- ```idxs``` and ```value_goal_idxs```, not the numeric values. If the index of the current state is the same as the index of the goal state, the agent gets successful surely.
> > > > >
> > > > > I may understand what you mean. The success is true only if the indexes of the states are the same. If the indexes are the same, the *state* and the *goal* must be equal. Thus, during relabelling, the reward is $1$ only if the goal is relabelled by the next state, i.e.,
> > > > >
> > > > > ```(state=s, action=a, next_state=s', reward=1, relabelled_goal=s').```
> > > > >
> > > > > Do you agree with this? BTW, could you confirm that, when evaluating the agent, if the reward is computed by the norm or the exact-match?

---

> > > > > > ### Author Response · Authors · 2025-08-04
> > > > > >
> > > > > > Thanks for your response.
> > > > > >
> > > > > > Yes, that’s exactly how goal relabeling is implemented in OGBench.
> > > > > >
> > > > > > During evaluation, success is determined based on the distance (i.e., the norm), rather than exact index matching. For example, in the maze environment, an episode is considered successful if the agent’s (x, y) position is within a small threshold of the goal position.
> > > > > >
> > > > > > This success condition in the evaluation logic can be found in the evaluation logic at line 442 of ``ogbench/locomaze/maze.py``:
> > > > > >
> > > > > > ``if np.linalg.norm(self.get_xy() - self.cur_goal_xy) <= self._goal_tol: info['success'] = 1.0``

---

> ### Comment · Reviewer_qrJ2 · 2025-08-04
>
> Thanks for your reply. I'm happy to discuss with you to clarify this issue. I'll update my score accordingly.
>
> Additionally, you don't need to add the experiments to compare the norm and the exact-match. It makes sense to use exact-match during goal-relabelling.

---

### Official Review · Reviewer_qyqg · 2025-06-05

**Clarity:** 3
**Significance:** 3
**Originality:** 3
**Rating:** 5
**Confidence:** 4

**Summary:**

The authors propose a new offline goal-conditioned RL (GCRL) algorithm called OTA. They begin by noting that the main bottleneck of HIQL (one of the state-of-the-art offline GCRL methods) is in the high-level policy. To address this, they propose training a separate high-level value function with temporally extended options (which are defined by subgoals), and extracting a high-level policy based on this high-level value function. They show that the resulting algorithm (OTA) substantially improves performance on many long-horizon navigation tasks in OGBench.

**Questions:**

I don't have additional questions other than the ones I asked in the weaknesses section.

**Ethical Concerns:**

["NO or VERY MINOR ethics concerns only"]

**Final Justification:**

My initial (mostly minor) concerns have been resolved, and I'd like to maintain the original score of 5 (accept).

**Limitations:**

Yes.

**Paper Formatting Concerns:**

I don't have any major formatting concerns.

**Quality:**

3

**Strengths And Weaknesses:**

[Strengths]
* The experiments in the motivation section (Section 4) are well designed. The authors convincingly argue that high-level policy learning is the bottleneck of HIQL, and OTA can resolve this issue effectively (Figure 5).
* The proposed method is simple and effective.
* The authors provide several analysis experiments (Sections 6.3-6.6), which are informative and helpful to understand the effect of the proposed method.
* OTA achieves substantially better performance than the previous methods on several challenging tasks (e.g., `humanoidmaze-giant` and `antmaze-*-explore`).
* The paper reads well in general.

[Weaknesses]

I don't see significant weaknesses of this work. Though I do have some concerns, mostly about empirical evaluation.
* From Table 4, OTA seems to require careful tuning of four hyperparameters ($\beta^h$, $\beta^\ell$, $k$, and $n$) for each setting. This potentially makes the empirical comparison unfair, given that the baselines use the same policy extraction hyperparameters for each dataset type (see Table 11 of the OGBench paper).
* The authors only selectively report performance on manipulation tasks. For example, how does OTA perform on the standard `play` datasets?
* OTA implicitly assumes that length-n trajectory chunks in the dataset are optimal (which is implicitly used in Eq. (4)). This is not always the case, especially when the dataset is highly suboptimal. The paper lacks a discussion about this point.

[Minor issues]
* L54: "by updating the value over a sequence of primitive actions" -- This sounds like the method does action chunking, which (as far as I understand) seems not to be the case.

---

> ### Author Rebuttal · Authors · 2025-07-31
>
> We sincerely thank Reviewer qyqg for the positive comments on the motivation of our paper and the analysis experiments. We also appreciate your constructive feedback regarding the empirical evaluation, including hyperparameters, manipulation tasks, etc. Below, we address your concerns in detail.
>
> ### Weaknesses 1: Hyperparameter tuning
>
> Thank you for the comments on tuning the hyperparameters of OTA. We agree that we searched the hyperparameters extensively to report our results. To show that OTA also outperforms HIQL with the same hyperparameters (e.g., $k, \beta^h, \beta^{\ell}$) as HIQL, we also carry out additional experiments in complex maze tasks (i.e., large and giant mazes except for explore dataset). The table below shows the results with the same hyperparameters to HIQL except for $n$. In this table, the success rate of OTA is still much higher than HIQL, meaning that simply applying the temporal abstraction can improve the performance.
>
>
> | Environment     | Type     | Size     | $n$   | $k$   | $\beta^h$  | $\beta^{\ell}$  | OTA | HIQL |
> |-----------------|----------|----------|-----|-----|-----|-----|-----------------|------------|
> | PointMaze       | navigate | large    | 5   | 25  | 3.0 | 3.0 | **85** ± 5 | 58 ± 5  |
> |                 |          | giant    | 5   | 25  | 3.0 | 3.0 | **72** ± 6 | 46 ± 9  |
> |                 | stitch   | large    | 5   | 25  | 3.0 | 3.0 | **46** ± 7 | 13 ± 6  |
> |                 |          | giant    | 5   | 25  | 3.0 | 3.0 | **44** ± 8 | 0 ± 0   |
> | AntMaze         | navigate | large    | 5   | 25  | 3.0 | 3.0 | **91** ± 1 | **91** ± 2  |
> |                 |          | giant    | 5   | 25  | 3.0 | 3.0 | **70** ± 2 | 65 ± 5  |
> |                 | stitch   | large    | 5   | 25  | 3.0 | 3.0 | **79** ± 3 | 67 ± 5  |
> |                 |          | giant    | 5   | 25  | 3.0 | 3.0 | **29** ± 5 | 2 ± 2   |
> |                 | explore  | medium   | 5   | 25  | 3.0 | 3.0 | **93** ± 3 | 37 ± 10 |
> |                 |          | large    | 10  | 25  | 3.0 | 3.0 | **62** ± 12 | 4 ± 5   |
> | HumanoidMaze    | navigate | large    | 20  | 100 | 3.0 | 3.0 | **82** ± 2 | 49 ± 4 |
> |                 |          | giant    | 20  | 100 | 3.0 | 3.0 | **91** ± 1 | 12 ± 4   |
> |                 | stitch   | large    | 20  | 100 | 3.0 | 3.0 | **43** ± 3 | 28 ± 3 |
> |                 |          | giant    | 20  | 100 | 3.0 | 3.0 | **61** ± 3 | 3 ± 2   |
>
>
> ### Weaknesses 2: Performance on play datasets
>
> To demonstrate the effectiveness of temporal abstraction between action sequences in the presence of noisy data, we conducted experiments using a noisy dataset. Additionally, we evaluated the performance of OTA on play datasets. The results are summarized in the table below. For a fair comparison, we fixed the hyperparameters $(k, \beta^h, \beta^{\ell})$ for both OTA and HIQL and varied only $n=2$.
>
> | Environment     | Data      |  Env |  OTA | HIQL |
> |-----------------|-----------|------|------|------|
> | Visual-cube     | play      | double | **51** ± 3| 48 ± 4|
> |        |     | triple  |**28** ± 1|21 ± 5|
> | Visual-scene    |    play | -  | **56** ± 5  | 50 ± 5 |
>
>
>
> ### Weaknesses 3: Assumptions on the data optimality
>
>
> First, we want to clarify that the update formula in Equation (4) does not assume data optimality. As in the definition of the option we used, we assume that the underlying policy is the behavior policy, which is used to collect the offline data, and the behavior policy is extremely suboptimal. Therefore, the option-aware value update in Equation (4) can be performed regardless of the data characteristics. In fact, we observed significant performance improvements on the extremely suboptimal datasets such as ``antmaze-medium-explore`` and ``antmaze-large-explore``.
>
>
> ### Minor issue: Option description
>
> Thank you for your comments on the notion of options we used. As the reviewer pointed out, we do not use action chunking, but the above explanation in the manuscript may induce misunderstanding. We will revise the explanation in the camera-ready version.

---

> > ### Comment · Reviewer_qyqg · 2025-07-31
> >
> > Thanks for the response. The new results with fixed hyperparameters look convincing to me. I think this is a good paper and would like to happily recommend acceptance.

---

### Official Review · Reviewer_x6Va · 2025-06-25

**Clarity:** 4
**Significance:** 3
**Originality:** 1
**Rating:** 4
**Confidence:** 4

**Summary:**

The paper investigates failure cases of HIQL (hierarchical implicit Q-learning) and proposes an improvement. The paper first provides a preliminary experiment that analyses failure cases of HIQL and concludes that these are probably caused by the high-level policy setting 'bad' goals, which in turn seems to come from imprecise V-value estimation. This problem is analyzed as being a problem mainly in long horizons. The proposed solution is to take a target value that instead of being based on 'regular' one-step bootstrapping, is based on effectively a n-step bootstrapping target. Experiments show that on various environments from OGBench and robot manipulation tasks, the proposed method outperforms non-hierarchical baselines as well as the original HIQL.

**Questions:**

Why are sign mismatches so crucial? To me it seems that going from 'just positive' to 'just negative' will change importance weight from slightly above 1 to slightly below one - might not be a big deal. Large errors that don't change the sign can make a bigger difference and more aversely affect the learned value function.

**Ethical Concerns:**

["NO or VERY MINOR ethics concerns only"]

**Final Justification:**

The author's replies have partially addressed my concerns. I have thus updated my overall rating. A few questions and concerns remain as communicated in my reply to the authors' rebuttal.

**Limitations:**

I think several limitations need to be more explicit:
- bias caused by going from 1-step to multi-step returns where the expectile function will now encourage a optimistic estimate of the whole n-steps
- assumption of deterministic dynamics.

**Quality:**

3

**Strengths And Weaknesses:**

Quality - Strenghts
- The paper provides a clear and well-reasoned analysis of failure cases of HIQL
- The empirical methodology over multiple environments is thorough and convincing. Results include error bounds over 8 random seeds.
- Beyond raw performance, a visualization is provided to confirm the intuition of why this method might work better, and an ablation is performed over different values of the 'n' parameter.

Quality - Weaknesses
- Conceptually and theoretically, the new method could have been analyzed in more detail. In more detail:
-- Limitations of the method are not mentioned. Some of these are inherited from IQL (sensitive to stochastic dynamics, which bias the expectile approximation of the 'max' operator over actions; recursive optimality), while some of them are increased by the method (the new target depends on (more) future states and actions, increasing the mentioned bias; the definition and analysis of "order inconsistency" seems incompatible with stochastic dynamics). To slightly expand on this point: the notion of "optimal trajectory' is not well defined, as in general an optimal policy in a stochastic environment can induce a distribution of trajectories. Due to non-anticipated ('noisy') transitions, "order consistency" should not be expected.
-- The paper strongly links the proposed approach to options, but the link to the original options in [46] is rather weak: in the original options paper, termination conditions are a Markov function (here: non-Markov), various options provide a high-level choice (here: only a single option), updates consider the aggretate discount gamma^n (here: n). In fact, I think the more useful analogy is to that of n-step returns.

Clarity - strengths
The paper is very clearly written in understandable language. Figures provide a clear illustration of the discussed ideas and contexts.

Clarity - weaknesses
As a rather minor weakness, I think the structure could be improved, as the introduction spends a lot of time detailing the proposed approach (better saved for the technical section) and is rather concise on the problem context, problem motivation, and solution need.

Significance
The empirical results are quite convincing, especially on the 'giant' versions of the mazes. On robot manipulation there is only a small (statistically likely non-significant) difference to HIQL, making it unclear how significant the result is beyond maps.

Originality
There is originality in the analysis of the problem, but the proposed solution has very limited originality, as this is essentially doing something very close to the well-known n-step returns.

Minor comments:
line 186: I'd rephrase this sentence. "noisy" is a bit ambiguous. I would typically interpret this as having a high variance, but the variance of the target doesn't depend on the horizon. Do you mean a bigger discrepancy between the target and the true value ('value error', if you will?).
Line 194: Options are clasically not considered the same as macro-actions, as macro-actions are open-loop and options can be closed loop. Macro actions would also typically be fixed-lenths while options classically have a Markovian termination condition.

---

> ### Author Rebuttal · Authors · 2025-07-31
>
> We sincerely thank Reviewer x6Va for the positive comments on our analysis of the failure cases of HIQL and the strength of our experimental results. We also appreciate your constructive feedback regarding the limitations in stochastic dynamics and the relationship to n-step TD learning, etc. Below, we address your concerns in detail.
> ### Weakness 1: Limitations inherited from IQL
> As the reviewer has mentioned, our method and analyses were developed under the assumption of a **deterministic environment**. We should have stated this point more clearly, especially when introducing concepts such as "order inconsistency" and the notion of an "optimal trajectory”, which only make sense in deterministic environments. We will revise the manuscript to clarify this assumption. While our method builds on IQL due to its simplicity and strong empirical performance, we do acknowledge that IQL has known limitations in stochastic environments.
>
> However, we note that this assumption is not unique to our work; several baseline methods [1,2,3] used in the offline GCRL benchmark also focus on deterministic environments, primarily due to the challenging nature of offline GCRL problems in more general stochastic environments. To that end, while our framework inherits certain limitations from IQL, we believe it still makes a significant contribution in deterministic environments by enabling objective comparisons with prior work under a consistent evaluation setting. Furthermore, we believe the core ideas of our approach (i.e., the option-aware target) could be extended to other offline RL algorithms that are more robust to stochastic environments [4,5], which we consider a promising direction for future work.
>
> As the reviewer pointed out, expectile loss may lead to over-optimistic value estimates in stochastic dynamics, using the $n$-step forward states for updating the value _may_ exacerbate the bias. However, in deterministic dynamics, our experiments show that multi-step returns reduce bias and improve performance, balancing the bias-variance trade-off.
> ### Weakness 2: Link to the original options
> - **Non-Markov termination condition**
>
> Thank you for pointing out that our use of a fixed n-step timeout is a non-Markovian termination condition. We would like to clarify that the original options framework by Sutton et al. [6] is general enough to accommodate such cases through the use of semi-Markov options.
>
> - **Options as high-level choices**
>
> While options can represent high-level choices when learned explicitly, our setting differs: we leverage them implicitly from the dataset. The behavior policy $\mu$ used during data collection consists of various high-level policies (e.g., grasping objects, moving items). Therefore, depending on which trajectories are sampled from the dataset, appropriate high-level choices are utilized, enabling temporal abstraction without explicit option learning.
>
> - **Aggregate discount $\gamma^n$**
>
> Several works, including the original options paper [6], apply a discount factor of $\gamma^n$ to rewards after $n$ steps. However, some studies [7,8,9] use option-level discounting when learning the value function. In our case, we adopt the option-level discounting scheme to enable temporal abstraction through an $n$-step target.
> ### Weakness 3: Connection with $n$-step TD learning
> The exact formula of TD target for $n$-step and OTA is as follows:
> - $n$-step target: $\sum_{i=0}^{n-1} - \gamma^i \cdot 1(s_i \neq g) + \gamma^n V_\theta (s_{t+n}, g)$
> - OTA target: $-1(s^{\Omega}\neq g) + \gamma V_\theta (s^{\Omega}, g)$
>
> In the above formula, since both targets mainly utilize n-step forward value, one _may_ think both formulas are quite similar. However, we want to clarify that extending the n-step target to the OTA target is not as straightforward. Rather, temporal abstraction through the option framework provides a natural explanation for the insights gained in our motivation experiments.
>
> Though the reward signal has a minor difference (i.e., the scale of rewards), the critical difference between $n$-step TD and OTA comes from the choice of $\gamma$, which controls the information decay ratio as we perform the TD update. Since the standard $n$-step target typically uses the same $\gamma$ as in 1-step target, the discount factor applied to the value function across the trajectory remains the same as in the 1-step TD. Or, equivalently, the discount factor applied to the value function in the n-step target decays exponentially with $n$ (e.g., $\gamma^n \approx 0.95$ for $(n,\gamma)=(5,0.99)$ or $(n,\gamma)=(10,0.995)$) while that in the OTA target is independent of $n$.
>
> To demonstrate that the excessive decay of the discount factor in the standard $n$-step target hinders learning of the order-consistent value, we carried out additional experiments comparing the order consistency ratio ($r^c$) of value functions learned by 1-step TD, $n$-step TD, and OTA, all using the same discount factor $\gamma$. We also used the same $n$ for $n$-step TD and OTA. In the table below, in contrast to OTA, we clearly observe that the $r^c$ for $n$-step TD has no improvement over 1-step TD since using the $n$-step TD target may still suffer from the estimation error of the value function due to the same level of discount factor.
>
> |env|$\gamma$|$n$|1-step TD ($n$=1)|$n$-step TD|OTA|
> |-|-|-|-|-|-|
> |antmaze-large-explore|0.99|15|75±1|77±1|**97**±1|
> |antmaze-giant-stitch|0.99|10|91±1|84±2|**94**±1|
> |humanoidmaze-large-stitch|0.995|20|75±1|76±0.3|**89**±3|
> |humanoidmaze-giant-stitch|0.995|20|72±1|72±0.3|**94**±1|
>
> The above results suggest that one may try to improve n-step TD by controlling $\gamma$ appropriately for each $n$ so that the discount factor can be adjusted. However, it is clear that such an approach would introduce additional complexity in hyperparameter selection. In contrast, our OTA fixes $\gamma$ regardless of $n$, which makes the scheme much simpler.
>
> In the revised manuscript, we will clarify the connection between OTA and $n$-step TD learning. In particular, we will address the *clarity feedback* by summarizing some of the detailed explanations in the Introduction and adding a discussion of the similarity to n-step TD in Section 5.
> ### Weakness 4: Small performance improvement in manipulation tasks
> We analyze the order consistency ratio ($r^c$) of $V^h$ in HIQL on ``cube-single-play`` and ``cube-double-play``, both achieving a high ratio of 0.98. The reason for achieving high $r^c$ in HIQL is the horizon of both tasks is relatively small (less than 500) compared to complex maze tasks. We, therefore, conclude that in the case of short-horizon manipulation tasks, the core problem lies in the policy extraction scheme for high-level policy (i.e., AWR for $\pi^h$), not in the value estimation. However, we believe that for more complex and long-horizon robot manipulation tasks in which 1-step TD cannot learn order-consistent value, OTA can improve the performance of HIQL.
>
> ### Minor comment: Clarification of "noisy" and "macro-action"
>
> Regarding line 186, we agree that the term "noisy" can be ambiguous. Here, it refers to high estimation error between predicted and true values. As described in HIQL (see Figure 8 in HIQL), the variance of the value function increases with longer horizons, destabilizing value learning and resulting in larger value errors, as illustrated in our Figure 2. We will clarify this explanation in the related work section and revise the wording.
>
> For line 194, we appreciate the distinction you pointed out between options and macro-actions. Since we define options along with the termination conditions from the offline dataset, they do not necessarily have fixed lengths. Therefore, the term "options" is more appropriate in our setting, and we will update the manuscript accordingly.
>
> ### Question 1: Importance of sign matching
>
> For high-level policy learning, the advantage is computed by $V^{h}(s_{t+k}, g) - V^{h}(s_t,g)$. With large $k$ values (e.g., 25 in ``antmaze``, 100 in ``humanoidmaze``), this results in large-magnitude advantages. If the sign of the advantage is incorrect, the magnitude of regression weights (which is the exponentiated advantage) is drastically increased or decreased, causing unintended improper subgoal regression for high-level policy. For example, an advantage range of $[-1, 1]$ with $\beta^h=3$, the regression weights vary from $e^{-3}\approx 0.05$ to $e^{3}\approx 20$, which is significantly far from 'just negative' or 'just positive'. In our experiment, the magnitude of the advantage often exceeds 10, and we also use $\beta^h>1$, making the impact of sign mismatch even more critical.
>
> While small-magnitude advantages may be less affected by sign errors, a low advantage magnitude implies that the objective becomes similar to behavior cloning, which does not give any useful information on learning proper subgoals. For this reason, we first scale $\beta^h$ to ensure sufficient advantage magnitude, and then apply AWR.
>
> Reference
>
> [1] Ghosh et al., “Learning to reach goals via iterated supervised learning”, ICLR 2021
>
> [2] Wang et al., “Optimal Goal-Reaching Reinforcement Learning via Quasimetric Learning”, ICML 2023
>
> [3] Park et al., “HIQL: Offline Goal-Conditioned RL with Latent States as Actions”, NeurIPS 2023
>
> [4] Villaflor et al., “Addressing optimism bias in sequence modeling for reinforcement learning”, ICML 2022
>
> [5] Liu et al., “A Tractable Inference Perspective of Offline RL”, NeurIPS 2024
>
> [6] Sutton et al., "Between MDPs and semi-MDPs: A framework for temporal abstraction in reinforcement learning.", Artificial intelligence 1999
>
> [7] Klissarov et al., "Flexible Option Learning", NeurIPS 2021
>
> [8] Nachum et al., "Data-Efficient Hierarchical Reinforcement Learning", NeurIPS 2018
>
> [9] Chai et al., "MA-RLHF: Reinforcement Learning from Human Feedback with Macro Actions", ICLR 2025

---

> > ### Author Response · Authors · 2025-08-04
> > **A gentle reminder for feedback**
> >
> > Dear Reviewer x6Va,
> >
> > Since the discussion period is almost approaching to its end, we would greatly appreciate your additional feedback on our response to your review. Would you please let us know whether our response has helped you further clarify our paper?
> >
> > Thank you very much in advance.

---

> > > ### Author Response · Authors · 2025-08-05
> > > **A gentle reminder for feedback**
> > >
> > > Dear Reviewer x6Va,
> > >
> > > Since the discussion period is almost approaching to its end, we would greatly appreciate your additional feedback on our response to your review. Would you please let us know whether our response has helped you further clarify our paper?
> > >
> > > Thank you very much in advance.

---

> > > > ### Author Response · Authors · 2025-08-06
> > > > **A gentle reminder for feedback**
> > > >
> > > > Dear Reviewer x6Va,
> > > >
> > > > As the discussion period is about to close, we would be grateful for any feedback, even a brief note, on our responses. Please let us know if our clarifications address your concerns or if further evidence is needed.
> > > >
> > > > Thank you for your time.

---

> > > > > ### Comment · Area_Chair_sdn6 · 2025-08-06
> > > > > **Post-rebuttal feedback**
> > > > >
> > > > > Dear reviewer x6Va,
> > > > >
> > > > > It would be appreciated if you can provide your input after the rebuttal from the authors, particularly given that you are the only reviewer with a score that tends towards rejection. Pease mention whether and why you keep your score.
> > > > >
> > > > > Best regards, AC

---

> > ### Comment · Reviewer_x6Va · 2025-08-08
> > **Thanks for the clarifications**
> >
> > First of all - my apologies for the delayed response. I had misunderstood the discussion period, meaning it took place during my holiday break during which I did not read my e-mails.
> >
> > Thank you for the clarifications.
> >
> > On limitations - I understand that there is value in studying 'special cases', such as deterministic MDPs, but I think it is important to communicate such limitations clearly. Similarly, thanks for clarifying that overall, with n-step you have lower bias, but I think it would be better to be upfront with this information.
> >
> > On the comparison to n-step returns: thank you for the clarification, this makes it more clear to me why the approach adds value.
> >
> > On sign mismatches - I am not completely convinced. Sure, if you have a big error in the advantage this will destroy your learning. But, then why not measure the size of the error? E.g., if I have (real, estimated) advantage pairs of (1.5, 0.5), (0,5, -0.5), (-0.5, -1.5) the weight calculated with the estimated advantage rather than the real advantage will be off by a factor exp(1) in all of these cases, whether there is a sign flip (as in the second example) or not (as in the first and third examples).

---

> > > ### Author Response · Authors · 2025-08-09
> > >
> > > We appreciate the constructive feedback and the opportunity for further discussion.
> > >
> > >
> > > First, we will clearly state the assumptions of the deterministic setting and acknowledge potential limitations, such as the overestimation introduced by n-step returns.
> > >
> > > Regarding sign mismatches, we estimate the advantage $V(s_{t+k}, g)-V(s_t, g)$ along near-optimal trajectories, where the true value is typically a large positive value (proportional to $k$). We also measured the proportion of cases where the absolute value of the estimated advantage is less than 1. For the ``antmaze-large-explore``, this proportion is about 5%, and for ``humanoidmaze-large-stitch``, it is about 7%. Therefore, the scenario we are concerned about—where both the true advantage and the estimated advantage have small magnitudes—is infrequent and has minimal impact on policy learning.
> > >
> > >
> > > In our case, the advantage is generally large, so a sign mismatch leads to severe errors that can significantly impair policy learning. While we acknowledge that large magnitude errors without a sign flip can also hinder policy learning, we believe these issues can be mitigated by tuning the parameter $\beta$ .
> > >
> > >
> > > In summary, sign-mismatch captures value errors that greatly impair policy learning, and our experiments demonstrate that the order-consistency ratio effectively indicates how much value learning deteriorates in long-horizon settings. We will also report the scale of the estimated advantage to further clarify the importance of sign mismatches.

---

### Official Review · Reviewer_1VvB · 2025-06-27

**Clarity:** 3
**Significance:** 3
**Originality:** 3
**Rating:** 4
**Confidence:** 4

**Summary:**

This paper focuses on the challenges of offline goal-conditional reinforcement learning (GCRL), pointing out that the failure of the hierarchical policy of HIQL is mainly due to the difficulty of generating suitable subgoals in the high-level policy, and the sign of the advantage signal frequently becomes incorrect. Thus, the paper proposes the Option aware Temporarily Abstracted (OTA) value learning method, which integrates temporal abstraction into temporal differential learning and improves the accuracy of advantage estimation. Experiments have shown that OTA performs well in complex tasks.

**Questions:**

1. Adjusting the temporal abstraction step of the value function in the high-level policy has improved the value estimation of long-horizon states, but does this have an impact on the value estimation of short-horizon states?
2. Would it be better to use separate value functions with different step sizes for the high and low level policy?
3. The temporal abstract step n and subgoal step k need to be adjusted in coordination. Is there any method that can more automatically determine the optimal combination of n and k instead of parameter tuning?

**Ethical Concerns:**

["NO or VERY MINOR ethics concerns only"]

**Limitations:**

yes

**Quality:**

3

**Strengths And Weaknesses:**

Strengths:\
This article first analyzes whether it is the bottleneck of the high-level policy or the low-level policy, then analyzes the reasons for the bottleneck of the high-level policy, and finally proposes methods. The whole process is gradual and easy to understand. The proposed method maintains simplicity. The experimental effect has been greatly improved.\
Weaknesses:\
Introduced a new temporal abstraction step hyperparameter n, increasing the complexity of parameter tuning. The method improves the consistency of the value function, but it cannot guarantee complete orderliness and has certain limitations.

---

> ### Author Rebuttal · Authors · 2025-07-31
>
> We sincerely thank Reviewer 1VvB for the positive comments on our analysis of the bottleneck in hierarchical policies and the simplicity of our method. We also appreciate your constructive feedback regarding the additional hyperparameter $n$ and the guarantee of complete orderliness, etc. Below, we address your concerns in detail.
>
> ### Weakness 1: Additional hyperparameter $n$
>
> We agree that introducing another hyperparameter $n$ may increase the complexity of parameter tuning. However, we would like to clarify that the search complexity of our method is not prohibitively large. Instead of performing an extensive hyperparameter sweep from scratch, you can follow the guidelines in Question 3 and attain strong performance within a much narrower search space. We hope this addresses your concern.
>
> ### Weakness 2: A limitation in guaranteeing complete order consistency
>
> We agree that our method does not fully guarantee perfect value ordering. Nevertheless, we would like to emphasize that our simple approach significantly reduces value‑ordering errors and, in turn, achieves much higher performance compared to the baselines.
>
> ### Question 1: Impact of step $n$ on short-horizon value estimation
>
> The objective of the high-level policy in long-horizon planning is to identify reachable subgoals that the low-level policy can achieve within a short-horizon. Accordingly, the high-level value function is only used for long-horizon planning. In contrast, training the low-level policy requires accurate value estimation within short horizons.
>
> That is, short-horizon value estimation is utilized solely for training the low-level policy through the *low-level value function*, whereas the temporal abstraction step $n$ affects only the *high-level value function*. Therefore, adjusting the temporal abstraction step $n$ turns out not to have a direct impact on the value estimation of short-horizon states.
>
> ### Question 2: Separate value functions with different step sizes for each policy
>
> We apologize for the lack of clarity in the manuscript and appreciate the reviewer’s attention to this point.
>
> Indeed, we agree with the reviewer and already follow this design in our approach. Specifically, unlike HIQL which trains a single value function $V$ for both $V^h$ and $V^\ell$, we train separate value functions for the high-level and low-level policies, each with a different step size -- i.e., the low-level value function, $V^\ell$, is trained using a standard 1-step target, while the high-level value function, $V^h$, uses an $n$-step option-aware target. We will further clarify this point in the final version of the paper.
>
>
> ### Question 3: Automatically selecting optimal $n$ and $k$
>
> While devising more automated strategies for selecting hyperparameters could be further explored, we have found that narrowing the search to a small, dataset‑ or environment‑specific range of plausible values works well in practice. For example, our OTA can perform effectively on short-horizon tasks such as AntMaze (≈ 1000 execution steps) with $(n,k) = (5,25)$, whereas it performs well on long-horizon tasks like HumanoidMaze (≈ 4000 execution steps) with $(n,k) = (20,100)$. This rough *linear scaling* trend of $(n,k)$ with respect to the execution steps implies that having prior knowledge about the specific execution steps of the dataset/environment can quickly narrow down the range of the relevant hyperparameters so that full exhaustive searches can be avoided.

---

> ### Comment · Reviewer_1VvB · 2025-08-04
>
> Thank you for the response. I will keep my score

---

### Note · Authors · 2025-08-12

We sincerely thank all reviewers for their constructive and insightful feedback during the rebuttal. The reviewers appreciated our experimental design in the motivation section that identifies the high-level policy as the bottleneck, the *"order consistency"* metric, and the *"option-aware temporally abstracted value"* as key strengths for long-horizon planning. Their insightful and constructive feedback improved our analyses and clarity.

- Reviewer 1VvB questioned about the the usage of additional hyperparameter $n$ and the automatic way to select the hyperparamters $(n,k)$. Those comments motivated us to further validate the role of $n$, and explore principled approaches for selecting the hyperparameter $(n,k)$.

- Reviewer x6Va raised concerns about OTA's susceptibility to overestimation in stochastic environments (same as IQL) and issue of sign-mismatch in advantages. We clarified the limitations and emphasized the importance of accounting for the magnitude of the advantage.

- Reviewer qyqg asked about the fairness of hyperparameter tuning and the assumptions regarding data optimality. We addressed these with fixed-hyperparameter experiments showing OTA's consistent performance and clarified that our method does not assume data optimality.

- Reviewer qrJ2 provided precise and constructive feedback on  terminology and the reward function implementation. This thoroughness helped us identify potential sources of misunderstanding and motivated us to refine our explanations and revise misleading terms.

All updates will be reflected in the paper to address the reviewer's points and further strengthen the empirical evaluation.

Due to timing constraints, we were unable to engage in a more in-depth final discussion with reviewer x6Va, but we hope our clarification on the sign-mismatch issue was helpful. To reiterate: (1) high-level advantage generally have a large scale, and (2) frequent sign-mismatches indicates a failure in long-horizon value learning. Therefore, *"order-consistency"* serves as an effective empirical indicator to evaluate how well the value function is learned. We will revise the manuscript to highlight this point by including an analysis of the estimated advantage scale.

---

### Decision · Program_Chairs · 2025-09-17

**Decision:**

Accept (spotlight)

**Comment:**

The paper provides a clear analysis of failure cases of HIQL. It then provides a simple, well-grounded and effective technique that outperforms existing baselines on long-horizon tasks. The experiments are quite thorough and also provide several interesting insights, such as about the accuracy of the estimated value function. All reviewers have scores of at least 4 (minor accepts). The few minor weaknesses that were pointed out relate to some remaining limitations and some additional research questions that are still open. However, all reviewers agree that the contribution is significant.